# One-hour extraction-free loop-mediated isothermal amplification HPV DNA assay for point-of-care testing in Maputo, Mozambique

Maria J. Barra [1], Alexis F. Wilkinson[1], Ariel E. Ma[1], Karthik Goli[1], Hira Atif[2,3,4], Nafissa M.R.B. Osman[2], Cesaltina Lorenzoni[2,3], Guilhermina Tivir[5], Eva H. Lathrop[6], Philip E. Castle[7], Ming Guo[8], Jane R. Montealegre[9], Ellen S. Baker[10], Mila P. Salcedo[10], Kathleen M. Schmeler[10] & Rebecca R. Richards-Kortum [1] ✉

Human papillomavirus (HPV) is responsible for nearly all cases of cervical cancer. Affordable point-of-care DNA testing is needed for cervical cancer screening in low- and middle-income countries, where most cervical cancer cases occur. HPV DNA testing typically requires complex lab infrastructure and trained personnel. In this work, we develop a loop-mediated isothermal amplification (LAMP)-based HPV DNA test, which targets three of the most oncogenic HPV types (HPV16, HPV18, HPV45) and a cellular control and achieves analytical sensitivity comparable to gold standard methods. Our extraction-free sample preparation strategy permits adding sample lysate directly to the LAMP reaction. We utilize a low-cost benchtop heater/fluorimeter, delivering results in less than one hour. We analytically evaluate our assay with clinical samples in Houston, Texas ($n = 38$) and Maputo, Mozambique ($n = 191$). Results show 100% and 93% concordance, respectively, with a reference test widely used in low-resource settings. This sensitive and specific four-step assay can potentially expand cervical cancer screening in resource-limited settings.

Accessible human papillomavirus (HPV) DNA testing is essential for effective cervical cancer screening, enabling the identification of individuals at high risk of developing the disease. Despite being highly preventable, the cervical cancer burden remains high, claiming approximately 350,000 lives in 2022[1]. Almost all cases result from persistent infection with high-risk HPV[2], which can be prevented through prophylactic vaccination and through screening, early detection, and treatment of precancerous lesions. The World Health Organization (WHO) recommends HPV DNA testing as the preferred method for cervical cancer screening[3] because of its high precision and sensitivity. However, the lack of affordable, point-of-care HPV tests has made it challenging to implement HPV DNA-based screening, particularly in low-resource settings.

[1]Department of Bioengineering, Rice University, Houston, TX, USA. [2]Universidade Eduardo Mondlane (UEM), Maputo, Mozambique. [3]Hospital Central de Maputo, Maputo, Mozambique. [4]Hospital Geral de Mavalane, Maputo, Mozambique. [5]Centro de Investigação e Treino em Saúde da Polana Caniço (CISPOC), Maputo, Mozambique. [6]Reproductive, Maternal, Newborn and Child Health, Jhpiego, Baltimore, MD, USA. [7]Division of Cancer Prevention, National Cancer Institute, Bethesda, MD, USA. [8]Department of Anatomical Pathology, The University of Texas MD Anderson Cancer Center, Houston, TX, USA. [9]Department of Behavioral Science, The University of Texas MD Anderson Cancer Center, Houston, TX, USA. [10]Department of Gynecologic Oncology & Reproductive Medicine, The University of Texas MD Anderson Cancer Center, Houston, TX, USA. ✉e-mail: rkortum@rice.edu

The majority of cervical cancer cases occur in low- and middle-income countries (LMICs), with the highest burdens of the disease in Sub-Saharan Africa[1]. LMICs face numerous challenges around the implementation of broad, population-level screening programs, including limited resources, inadequate infrastructure, and a shortage of trained personnel[4]. Consequently, in many LMICs, only a small proportion of the population is screened. Even when screening programs are in place, results are often not available until after the initial visit, and patients who test positive often miss out on necessary treatment due to numerous structural and individual-level barriers. A rapid, accurate HPV test that could be performed at the point of care could enable a screen-and-treat approach, allowing patients to be screened and, if HPV-positive, undergo treatment in a single visit, as recommended by the WHO[3].

In late 2024, the WHO published a target product profile (TPP) for point-of-care HPV screening tests[5], specifying minimum and preferred characteristics. Tests that meet the TPP requirements are needed to meet the WHO's goal of screening 70% of women worldwide twice in their life between ages 35 and 45 years by the year 2030. These tests are especially needed in low-resource settings. To meet the TPP requirements, tests should be able to detect at least eight high-risk HPV types that cause ~97% of cervical cancer cases. Tests should include an internal quality control, temperature-stable reagents, and should not cross-react with other microorganisms that can be found in the genital tract. Additionally, tests should include all consumables in a kit that is easy to use and minimizes cost and specimen handling.

While significant advances have been made to improve HPV DNA test accessibility, inherent limitations of available tests have precluded widespread implementation in LMICs outside of research studies. Some of these limitations do not meet the TPP requirements, including, for example, cost, number of user steps, and infrastructure required. Thus, research is needed to design tests to address these limitations. Some available HPV DNA tests that have been implemented in research studies in LMICs include careHPV (Qiagen, Germany)[4,6,7], Cobas (Roche, Switzerland)[8,9], Onclarity (BD, USA)[8], ScreenFire (Atila Biosystems, USA)[10,11], and GeneXpert (Cepheid, USA)[10,12,13]. GeneXpert HPV is a highly sensitive polymerase chain reaction (PCR)-based test; GeneXpert is one of four HPV tests prequalified by the WHO and is the only prequalified test that is suitable for use near the point of care. PCR is sensitive and specific and is the traditional method for DNA testing. GeneXpert can detect 14 HPV types, contains all necessary reagents in a sealed cartridge, and requires minimal user steps. A sample is added to the cartridge, which is placed into the GeneXpert instrument for cell lysis through sonication, DNA purification, and PCR amplification[14]. The assay has a sample-to-answer time of one hour. Despite its ease of use, GeneXpert's major limitations are its high per-test cost of approximately USD $15 and a minimum instrument cost of USD $9,420[15], both of which are barriers to uptake in LMICs.

In this work, we describe the development and evaluation of an extraction-free sample-to-answer loop-mediated isothermal amplification (LAMP) assay designed to detect the top three HPV types attributable to cervical cancer (HPV16, HPV18, and HPV45), which cause 75% of cases[16], along with a cellular control to ensure sample adequacy. Specifically, we employ a LAMP method previously developed by Tanner et. al.[17] called "Detection of Amplification by Release of Quenching" (DARQ) LAMP. LAMP offers several advantages over PCR, including faster amplification at a single temperature, even in the presence of inhibitors commonly found in clinical samples[18]. These benefits allow for a simpler workflow and faster results, eliminating the need for complex equipment[19]. Our workflow minimizes specimen handling by incorporating extraction-free sample preparation, in which we chemically lyse cervicovaginal swab samples and add lysate directly to LAMP reagents. The simplicity of this step is a key factor that allows the assay to be performed at the point of care in resource-limited settings. Our results demonstrate that LAMP can be effectively performed directly on these lysates with minimal loss of assay sensitivity. We implement our four-step assay on a compact and low-cost benchtop heater and fluorimeter (T8-ISO, Axxin Lty., Australia).

In addition to assay development, we present an evaluation of the assay's analytic performance. We collect and test clinical samples in a high-resource (Houston, TX) and a low-resource setting (Maputo, Mozambique). Mozambique is an LMIC with one of the highest burdens of cervical cancer in the world[20]. Similar to many LMICs, it is estimated that less than 5% of women in Mozambique have been screened for cervical cancer[21]. A rapid point-of-care test deployable in low-resource settings such as those in Mozambique could facilitate large-scale screen-and-treat strategies in a single visit. Our sample-to-answer HPV LAMP assay, which requires less than one hour, is sensitive, specific, easy to perform, and requires minimal equipment is a proof of concept that has the potential to fill this need. Here, we describe the development and demonstrate the analytic performance of a protype assay for the three most carcinogenic HPV types as a first step in fulfilling WHO's TPP for a HPV test.

## Results

For this work, we used the DARQ LAMP method depicted in Fig. 1. A primer set, consisting of six primers (F3, B3, forward inner primer (FIP), backward inner primer (BIP), loop forward (LF), and loop backward (LB)), is used for each DNA target. One of the LAMP primers, in this case FIP, is labeled with a quencher. FIP consists of two contiguous sequences: F1c and F2. A probe (Fd) with a sequence complementary to F1c is labeled with the corresponding fluorophore and bound to F1c. When F1c (with quencher) and Fd (with fluorophore) are bound, the fluorophore is quenched. The F2 sequence of the FIP primer binds to its target and amplification proceeds, generating a new strand with the F1c:Fd duplex. When a different primer binds to this strand and amplification occurs, Fd is displaced and the fluorophore and quencher are separated enabling fluorescence emission. This process occurs exponentially, producing proportional fluorescence.

### Singleplex DARQ LAMP assays

We began by designing singleplex DARQ LAMP assays for each target based on our previous work[22]. In all assay development experiments, we used DNA extracted from HPV-infected human cell lines as the target (SiHa – HPV16, HeLa – HPV18, MS751 – HPV45, Jurkat – human genomic DNA (gDNA)). We first evaluated the performance of the singleplex DARQ LAMP assays using a Bio-Rad thermocycler (Fig. 2, Supplementary Fig. 1). For HPV16, all three replicates of 50 HPV16 copies/reaction amplified in approximately 20 min, and only one replicate of 10 input copies amplified (Fig. 2A). In the gDNA assay, 1 ng of gDNA consistently amplified in under 45 minutes, but no replicates of 0.1 ng/reaction amplified (Fig. 2B). These results were considered satisfactory to move forward with multiplexing both reactions by simply adding the primer mixes for both targets in the same reaction, with HPV16 amplification being read on the FAM channel and gDNA amplification on the ROX channel.

For HPV18, 10 input copies amplified in less than 40 minutes, but reactions with a single input copy amplified inconsistently (Fig. 2C). The HPV45 assay performed less well using this strategy. All replicates of 100 HPV45 copies/reaction amplified consistently in under 30 min, but only one replicate each of 10 and 50 input copies amplified, indicating that this assay was less sensitive and more inconsistent (Fig. 2D). To multiplex the HPV18 and HPV45 assays, we used a non-specific DNA-intercalating dye instead of DARQ LAMP for amplicon detection to improve sensitivity. Although this method does not allow for genotyping, infections with both HPV types are managed identically in a screen-and-treat approach according to the WHO's guidelines[3]. Additionally, of the previously mentioned commercially available tests, careHPV, ScreenFire, and GeneXpert do not differentiate HPV18 and HPV45. Therefore,

Forward inner primer (FIP)

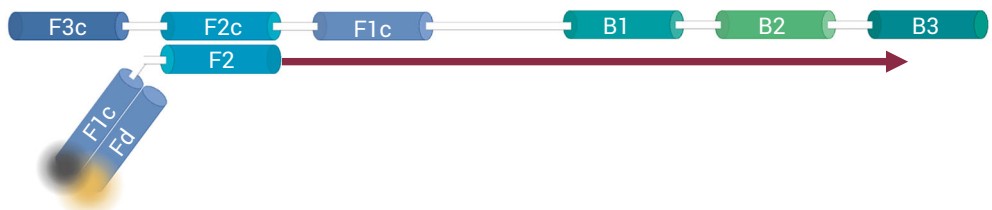

1. Labeled FIP:Fd duplex binds to template (HPV16, 18, 45 or gDNA) and is extended. Fluorophore is quenched.

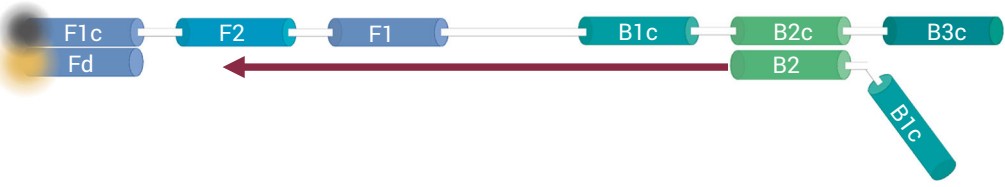

2. BIP primer binds to opposite end of newly synthesized labeled template and is extended.

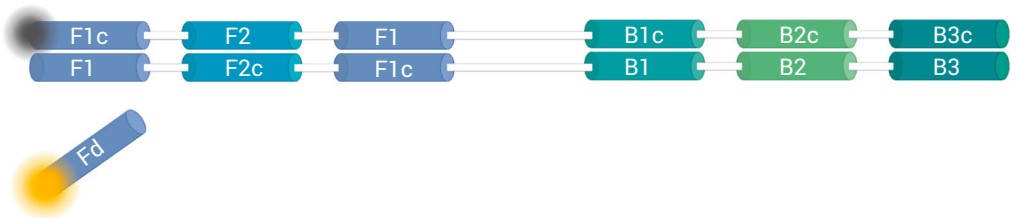

3. Fd is displaced during synthesis, separating quencher and fluorophore and enabling fluorescence emission.

**Fig. 1 | Representation of DARQ LAMP.** The FIP primer is labeled with a quencher and bound to a probe (Fd) with the corresponding fluorophore. FIP binds to its target DNA, extension occurs, and a new, labeled copy of target DNA is produced. Different unlabeled primers bind to this copy and produce extension, displacing the probe from the F1c sequence. Because the fluorophore and quencher become separated, fluorescence is produced as amplification occurs. Adapted with permission from Tanner et. al.[17]. Created in BioRender. Barra, M. (2025) https://BioRender.com/6tffa8x.

this decision also enables a direct comparison of our results with GeneXpert, which was used as the reference test in this work.

### Multiplexed HPV16/gDNA reaction

We optimized the multiplexed HPV16/gDNA DARQ LAMP reaction (Supplementary Figs. 2–6) and tested it with a range of input copies on the Bio-Rad thermocycler to conduct probit analysis (Fig. 3A, B, Supplementary Fig. 7A, B). The reaction was run for 45 min to meet our goal sample-to-answer time of less than one hour. With 95% probability, the limits of detection of the multiplexed assay were 57 HPV16 copies/reaction and 0.167 ng gDNA/reaction. For gDNA as the target, the optimized multiplexed reaction was faster and more sensitive than the original singleplex reaction (Fig. 2B). We also tested the assay on our target device, the Axxin T8-ISO, a portable, low-cost benchtop heater/fluorimeter that can be used at the point of care (Fig. 3C, D, Supplementary Fig. 7C, D). As expected, similar performance was observed on this device as with the Bio-Rad.

### Multiplexed HPV18/45 reaction

Similarly to the previous reaction, we optimized the multiplexed HPV18/45 reaction (Supplementary Fig. 8) and tested it with a range of input copies on the Bio-Rad thermocycler to conduct probit analysis (Fig. 4A, B, Supplementary Fig. 9A, B). With 95% probability, the limits of detection of the multiplexed assay were 11.7 HPV18 copies/reaction and 14.3 HPV45 copies/reaction. We also ran the assay on the T8-ISO (Fig. 4C, D) and obtained similar results demonstrating that the assay is fast and sensitive. Overall, differences in time to detection between the two devices were minimal and likely attributed to the stochastic nature of amplification initiation rather than any differences between devices. These results indicate that the low-cost, portable T8-ISO is an appropriate device to run this assay

### Analytic specificity of the multiplexed assays

The specificity of both reactions was tested with plasmids containing the full genomes of 13 off-target HPV types developed by the Centers for Disease Control and Prevention[23] (Supplementary Fig. 10). Additionally, a background of gDNA was added to each sample to more realistically represent clinical samples. As expected, in the HPV16/gDNA assay (Supplementary Fig. 10A) all three replicates of the on-target HPV16 plasmid amplified at 10 min. All the samples produced amplification in the ROX channel due to the background of gDNA. There were six replicates of gDNA alone with no plasmid, which

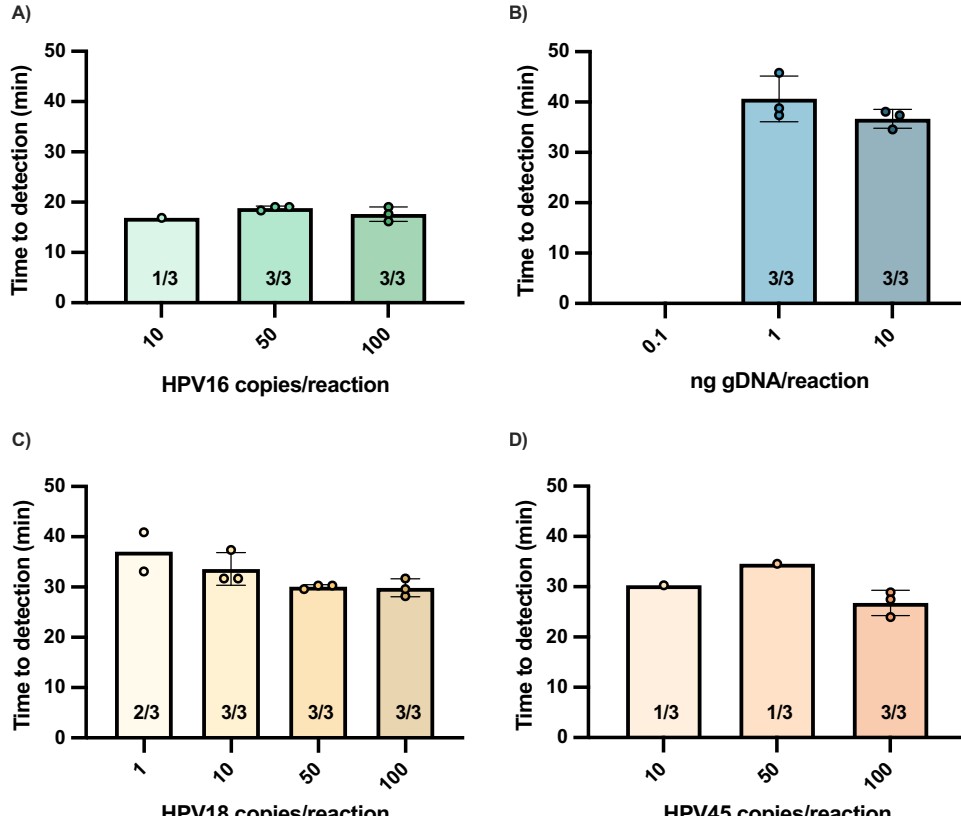

**Fig. 2 | Time to detection of a range of input copies of target in singleplex DARQ LAMP reactions performed on the Bio-Rad thermocycler.** The reactions are for **A** HPV16, **B** gDNA, **C** HPV18, and **D** HPV45. Bar graphs represent mean time to detection and error bars represent standard deviation. Number of replicates that amplified out of total replicates performed are specified on the bar graphs. Source data are provided as a Source Data file.

generally amplified earlier than samples that included plasmids. Interestingly, HPV16 samples showed the latest amplification on the ROX channel, likely due to competing reagents and preference for HPV16 amplification over gDNA amplification when the HPV16 target is in excess. In the HPV18/45 reaction (Supplementary Fig. 10B), only the HPV18 and HPV45 plasmids amplified as expected, both at ~10 min. No off-target amplification was observed, demonstrating targeted HPV type fidelity.

## Limit of detection with extraction-free crude lysate from cultured cells

To simplify the assay workflow for use at the point of care, we next incorporated extraction-free sample preparation. This is a crucial step in making the assay applicable at the point of care and in resource-limited settings, as it minimizes sample handling and does not require additional equipment. An illustration of the four-step workflow is shown in Supplementary Fig. 11. As a proxy for clinical samples, we evaluated the limit of detection of the sample-to-answer LAMP assay on the Bio-Rad with HPV-infected cultured cells (SiHa, HeLa and MS751) mixed with a background of non-HPV-infected cervical cells (C33A).

The chosen lysis method was sodium hydroxide (NaOH) which produces OH⁻ ions that break fatty acid-glycerol ester bonds in the cell membrane[24]. First, we evaluated the effects of a range of concentrations of NaOH (0 mM–100 mM) in the HPV16/gDNA reaction to analyze if it would be inhibitory to DNA amplification (Supplementary Fig. 12A). We observed a slight increase in time to detection for reactions containing between 2.5 and 5 mM NaOH and a significant increase for reactions containing between 5 and 10 mM NaOH. At 25 mM NaOH, amplification was completely inhibited. We proceeded to lyse cells using 1 M NaOH, which translated to a 4 mM NaOH concentration when lysate was directly added to the LAMP reaction.

Figure 5 shows results when lysate was directly added to the LAMP reaction. In the HPV16/gDNA reaction, samples containing 50 or more SiHa cells consistently amplified on the HPV16 fluorescence channel (FAM) at about 20 min (Fig. 5A). For samples containing 25 SiHa cells, 7/10 replicates amplified, and probit analysis determined that the limit of detection with 95% probability was 30.4 SiHa cells/reaction. The same samples were also read on the gDNA fluorescence channel (ROX) (Fig. 5B). Because all samples were in a background of C33A cells and SiHa cells also include gDNA, all replicates for all samples amplified as expected.

In the HPV18/45 reaction, 5/10 and 8/10 replicates amplified for samples containing one and five HeLa cells/reaction, respectively, and all replicates with higher cell counts amplified (Fig. 5C). This is expected, as HeLa cells each contain 10–50 copies of HPV18 DNA[25]. Additionally, when using small sample volumes with such low cell counts, it is possible that the sample volume may not contain any cells. Probit analysis showed that the limit of detection is 7.6 HeLa cells/reaction with 95% probability. For MS751, all replicates for the lowest cell count analyzed (10 cells/reaction) amplified in under 40 min (Fig. 5D), so probit analysis to determine the limit of detection could not be performed. However, the time to detection with 10 cells/reaction was considerably more variable than that for samples with 25–75 cells/reaction, suggesting that this is close to the limit of detection. These experiments show that both multiplexed assays are highly sensitive with the extraction-free sample-to-answer workflow.

## Sample-to-answer LAMP test with clinical samples in Houston, Texas

Next, we evaluated the sample-to-answer LAMP test with cervicovaginal swab samples collected into phosphate buffered saline (PBS) or nuclease-free water from patients attending colposcopy clinic follow-up visits in Houston, Texas. First, we evaluated if there was any

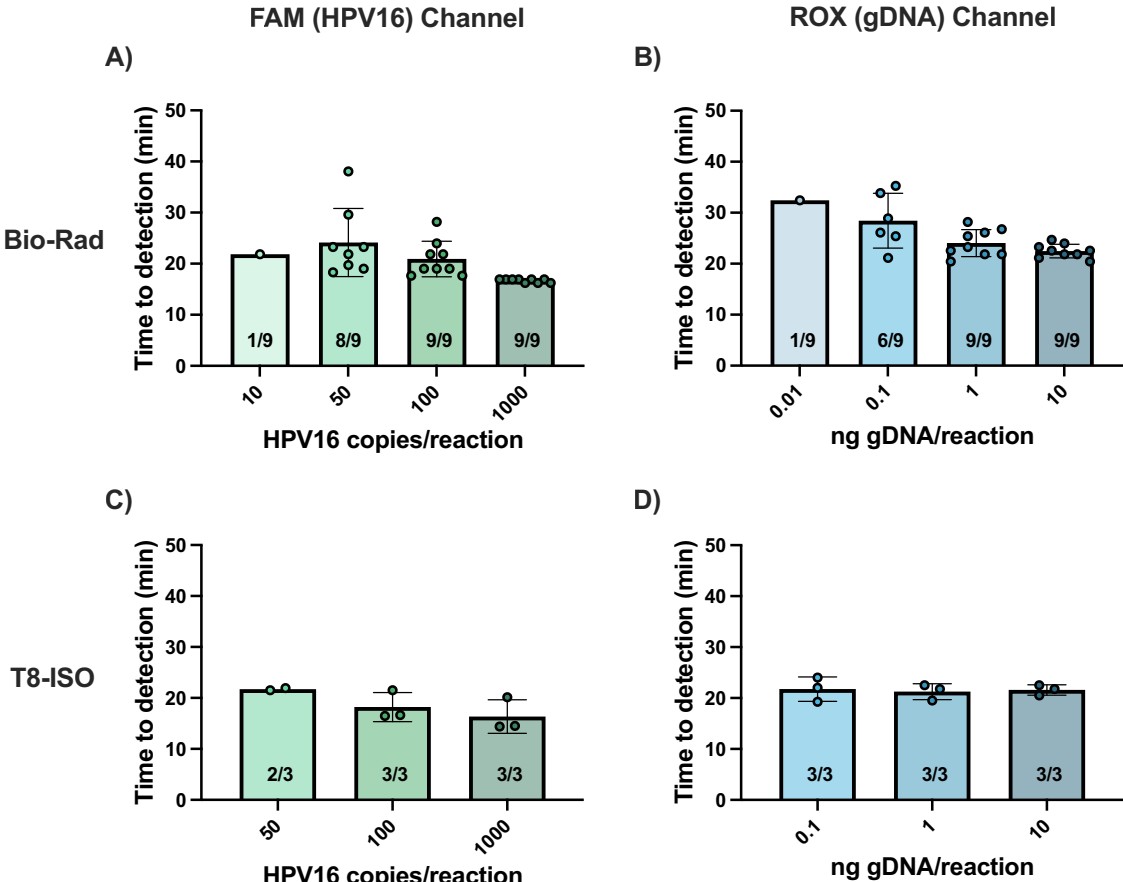

**Fig. 3 | Time to detection of optimized multiplexed HPV16/gDNA reaction on Bio-Rad thermocycler and benchtop T8-ISO.** A range of concentrations of SiHa (HPV16) DNA and gDNA were used as targets to evaluate the limit of detection of the reaction. The reaction was performed on the Bio-Rad (**A**, **B**) for probit analysis (*n* = 9) and was read on the **A** FAM channel for HPV16 detection and **B** ROX channel for gDNA detection. The reaction was also performed on the benchtop T8-ISO (**C**, **D**), also read on the **C** FAM channel for HPV16 detection and **D** ROX channel for gDNA detection (*n* = 3). Bar graphs represent mean time to detection and error bars represent standard deviation. Number of replicates that amplified out of total replicates performed are specified on the bar graphs. Source data are provided as a Source Data file.

impact of PBS on the HPV16/gDNA assay efficiency within a range of 0–14% v/v PBS in the amplification reaction (Supplementary Fig. 12B). There were no statistically significant differences in time to detection within this range, and replicates for each condition were consistent amongst each other. Adding lysate directly to the amplification reactions resulted in concentrations of 4% v/v PBS and 4 mM NaOH in this sample-to-answer testing.

We compared results of the LAMP test to those from a reference standard of GeneXpert HPV. By the reference standard, five samples were positive for HPV16, four were positive for HPV18/45, five samples were positive for other high-risk HPV types, and 24 were HPV-negative. For testing with LAMP, we used the four-step workflow previously described (Supplementary Fig. 11). 1 M NaOH was added to each sample and incubated at room temperature for 10 min, and the lysate was added directly to the LAMP amplification mixture. The results of the LAMP assay agreed with the reference standard with perfect concordance for all 38 samples (Table 1). Additionally, the gDNA cellular control amplified for all samples. Histologic analysis was performed on a subset of the Houston samples which can be found in Supplementary Tables 1 and 2; all four samples from women with cervical or vulvar precancer tested positive by the LAMP assay.

### Sample-to-answer LAMP test with clinical samples in Maputo, Mozambique

Our next objective was to evaluate the performance of the LAMP test in a low-resource setting with banked cervicovaginal swab samples.

These samples had previously been collected into PreservCyt storage buffer, so we first analyzed the impact of PreservCyt on the HPV16/gDNA assay (Supplementary Fig. 12C). Time to detection started to increase at ≥6% v/v PreservCyt/reaction, so we used 4% v/v PreservCyt/reaction and 4 mM NaOH in this sample-to-answer workflow.

The same four-step workflow (Supplementary Fig. 11) was used to perform the LAMP assay for banked samples in Maputo, Mozambique, and results were compared to the GeneXpert HPV reference standard (Table 2). Supplementary Fig. 13 shows representative amplification curves from the clinical samples, and Supplementary Fig. 14 shows representative photographs of the clinical samples, illustrating the variability in sample composition and complexity. Ninety-one percent (39/43) of samples that were only HPV16 positive by GeneXpert were also positive on the HPV16 LAMP test. One of these samples was additionally positive for HPV18/45 on the LAMP test but negative for HPV18/45 on the GeneXpert test. Eighty-eight percent (58/66) of samples that were only HPV18/45 positive by GeneXpert were positive on the LAMP HPV18/45 assay. All samples that were only positive for HPV18/45 on the LAMP assay were HPV18/45 positive on GeneXpert. Eight samples were positive for HPV16 and HPV18/45 by GeneXpert; of these, five were positive for both groups on the LAMP test, two were positive for just HPV18/45 and one was HPV negative on LAMP. Seventy-four samples were negative for HPV16/18/45 on GeneXpert; 36 were HPV-negative and 38 were positive for other high-risk HPV types. Ninety-nine percent (73/74) of these samples were also negative on the LAMP test. There was one sample which amplified on the HPV18/45

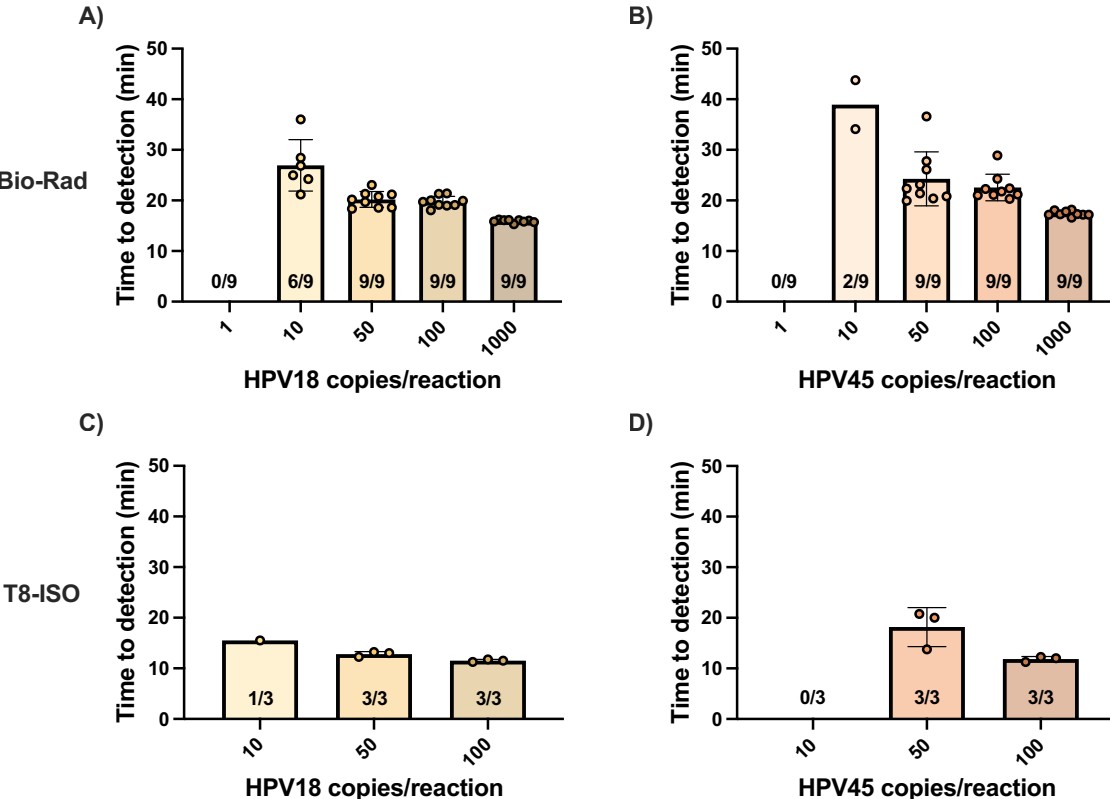

**Fig. 4 | Time to detection of optimized multiplexed HPV18/45 reaction on Bio-Rad thermocycler and benchtop T8-ISO.** A range of concentrations of HeLa (HPV18) and MS751 (HPV45) DNA were used as targets to evaluate the limit of detection of the reaction. The reaction was performed on the Bio-Rad (**A**, **B**) for probit analysis (*n* = 9) and was read on the FAM channel with **A** HPV18 target and **B** HPV45 target. The reaction was also performed on the T8-ISO (**C**, **D**) and read on the FAM channel with **C** HPV18 target and **D** HPV45 target (*n* = 3). Bar graphs represent mean time to detection and error bars represent standard deviation. Number of replicates that amplified out of total replicates performed are specified on the bar graphs. Source data are provided as a Source Data file.

LAMP assay but not on GeneXpert. For all the samples except one, the gDNA cellular control was positive in the LAMP assay. The exception was a sample that was HPV16-positive on both GeneXpert and LAMP and was considered a valid result.

Both false positive LAMP results were from samples that were positive for multiple other high-risk HPV types on GeneXpert. The HPV16 positive sample that was falsely positive for HPV18/45 on the LAMP test was positive for the P1 (HPV16), P3 (HPV31/33/35/52/58), P4 (HPV51/59), and P5 (HPV39/56/66/68) channels on GeneXpert, indicating infection with at least four high-risk HPV types. The sample that was negative for HPV16/18/45 on GeneXpert and falsely positive for HPV18/45 on the LAMP test was positive for the P3, P4, and P5 channels on GeneXpert, indicating infection with at least three high-risk HPV types. Additionally, we evaluated the cycle threshold (Ct) values for each of these channels. Samples with lower Ct values likely have more HPV DNA available for amplification. For the first and second sample the Ct value for every channel was <25 and <33, respectively, indicating very high viral loads of each HPV type. This may have contributed to off-target amplification and a false positive result in the HPV18/45 LAMP assay.

Positive test concordance between LAMP and GeneXpert was defined on the basis of clinical relevance, as patients who test positive for HPV16 and/or HPV18/45 undergo the same clinical management in a screen-and-treat approach according to WHO guidelines[3]. Thus, positive test results were considered concordant if a sample that was positive for HPV16 and/or HPV18/45 on GeneXpert was positive for at least one of those HPV types on the LAMP test. For the Mozambique samples, there was 93% concordance (177/191) (95% CI: 88-96%) between the two tests with a very good kappa value (kappa = 0.851,

95% CI: 0.777–0.926). There was 96% concordance for HPV16 (184/191) (95% CI: 93–99%), which had the highest kappa value (kappa = 0.902, 95% CI: 0.831–0.973). For HPV18/45 there was 94% concordance (180/191) (95% CI: 90–97%) with a kappa value of 0.877 (95% CI: 0.806–0.947).

Combining results from the two studies, there was 94% concordance (215/229) (95% CI: 90–97%) with a kappa of 0.878 (95% CI: 0.816–0.939) between the two tests. Among the 126 specimens that tested positive for HPV16, HPV18, and/or HPV45 by GeneXpert, 113 (89.7%) tested positive by the LAMP assay (analytic sensitivity relative to GeneXpert).

For the samples from Mozambique, we compared the GeneXpert Ct values for GeneXpert positive samples that were concordant by LAMP (true positive) or discordant by LAMP (false negative). The average GeneXpert Ct values for samples that were falsely negative by LAMP were significantly higher than those of the true positive samples for both the HPV16/gDNA assay and the HPV18/45 assay (Fig. 6A). This indicates that the samples that were falsely negative on the LAMP test generally had less HPV DNA available for amplification.

Next, we evaluated the relationship between time to detection (TTD) on the LAMP test and Ct value on GeneXpert. In the HPV16/gDNA reaction, the TTD of all samples with GeneXpert Ct < 30 (more HPV DNA) ranged from 14.2 to 29.8 min, whereas samples with GeneXpert Ct > 30 (less HPV DNA) ranged from 16.4 to 40.7 min (Fig. 6B). In the HPV18/45 reaction, all samples with Ct < 29.9 were detectable on the LAMP test in under 20 min (Fig. 6C). At Ct values > 29.9, there was no clear relationship, with time to detection ranging from 10.8 to 41.8 min. These data show that samples with Ct values greater than 30 on GeneXpert generally took longer to become detectable on the LAMP assay.

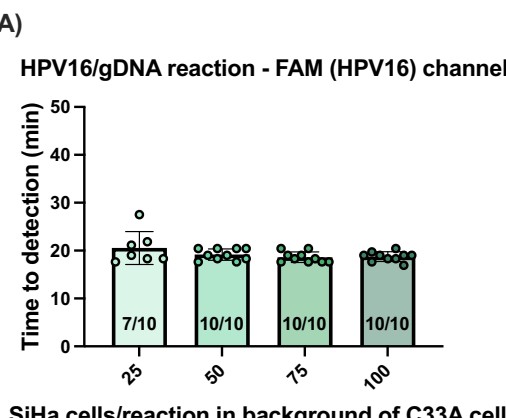

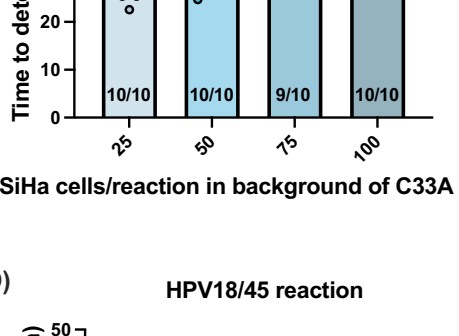

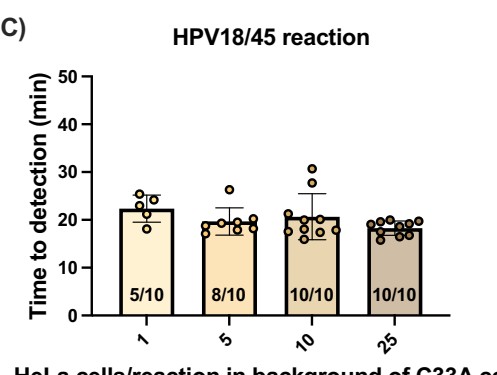

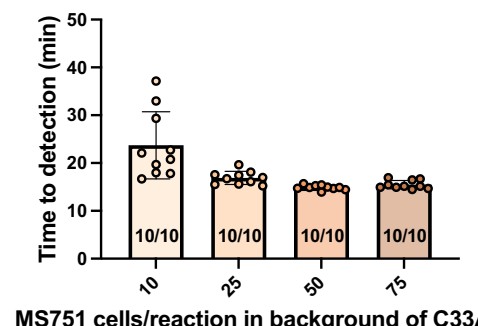

**Fig. 5 | Limit of detection of the multiplexed reactions with extraction-free sample-to-answer workflow using cultured cells on the Bio-Rad thermocycler.** A range of SiHa cells in a background of -10³ C33A cells/reaction were lysed for 10 minutes and immediately added to the HPV16/gDNA reaction. **A** Results of the FAM channel detecting HPV16 amplification and **B** ROX channel detecting gDNA amplification. Similarly, a range of **C** HeLa or **D**) MS751 cells in a background of -10³ C33A cells/reaction were lysed for 10 min and added to the HPV18/45 reaction. Bar graphs represent mean time to detection and error bars represent standard deviation. Number of replicates that amplified out of total replicates performed are specified on the bar graphs. Source data are provided as a Source Data file.

## Discussion

In this work, we developed, optimized, and evaluated a point-of-care extraction-free sample-to-answer LAMP test for HPV16, HPV18, HPV45 DNA, and a cellular control. We minimized user steps by incorporating extraction-free sample preparation, adding the sample lysate directly to the LAMP reagents. The assay was incubated, and fluorescence signal was detected on a simple benchtop fluorimeter (Axxin T8-ISO). The entire sample-to-answer workflow takes less than one hour, comparable to the reference test, GeneXpert HPV, which takes one hour.

Throughout this work, we tested our reaction on two different instruments - a high complexity thermocycler (Bio-Rad CFX Opus 96 Real-Time PCR System) and a simple fluorimeter/heat block (Axxin T8-ISO) and observed no significant differences in results obtained between the two. We optimized the assays on the Bio-Rad, which provides higher throughput, allowing us to test various conditions simultaneously and implemented the assays with clinical samples on the T8-ISO. The Bio-Rad is large ($33 \times 56 \times 36$ cm and 22 kg), expensive (-USD $25,000), and requires an electrical outlet, making it more difficult to implement in resource-limited settings. In contrast, the T8-ISO produces the same results in a low-throughput format, is portable ($18.5 \times 20.3 \times 11.0$ cm and 2.3 kg), relatively inexpensive (-USD $6,500), and can be run on a battery, making it ideal for resource-limited settings. Although we present results only on these two devices, a variety of mini thermocyclers and benchtop fluorescence isothermal readers are available on the market that could be used at the point of care. We envision this

LAMP assay to be compatible with a wide assortment of devices, expanding accessibility.

We evaluated the limits of detection of the LAMP assay with two different sample types: (1) purified DNA extracted from cultured cell lines and (2) our complete sample-to-answer workflow with whole cultured cells. With extracted DNA we obtained sensitivities of 57, 11.7, and 14.3 copies/reaction for HPV16, HPV18, and HPV45, respectively. These limits of detection are higher than those reported by GeneXpert, which detects 10 copies/reaction for all three HPV types[26]. However, with our extraction-free sample-to-answer workflow, the limit of detection compared favorably to that of GeneXpert. For the LAMP HPV16, HPV18, and HPV45 assays the limits of detection were 30.4 SiHa, 11.7 HeLa, and <10 MS751 cells/reaction, respectively, whereas those reported by GeneXpert are 71 SiHa, 46 HeLa, and 180 MS751 cells/reaction[26]. While these experiments are necessary, they do not take into consideration the complexity of real patient samples, which is why testing with clinical samples is necessary to evaluate assay performance.

In evaluating the LAMP HPV assay with cervicovaginal swab samples, we found 100% and 93% agreement between our test and GeneXpert HPV in Houston (38 samples) and Maputo (191 samples), respectively. Although a smaller number of samples were collected in Houston, the differences in the sample collection process at both sites are noteworthy. For example, the Mozambique samples had previously been collected into 20 mL PreservCyt media, 1 mL was tested on GeneXpert, the remaining sample was centrifuged and resuspended in approximately 1 mL of PreservCyt and then stored for up to

**Table 1 | Clinical sample test results on the HPV LAMP test versus GeneXpert in Houston, Texas. Gray colored cells highlight concordance for the paired results**

| LAMP HPV | GeneXpert HPV (Reference Test) | | | | |
| --- | --- | --- | --- | --- | --- |
| | HPV16 | HPV18/45 | HPV16 & HPV18/45 | HPV16/18/45 negative | Total |
| HPV16 | 5 | 0 | 0 | 0 | 5 |
| HPV18/45 | 0 | 4 | 0 | 0 | 4 |
| HPV16 & HPV18/45 | 0 | 0 | 0 | 0 | 0 |
| HPV16/18/45 negative | 0 | 0 | 0 | 29 | 29 |
| Total | 5 | 4 | 0 | 29 | 38 |

**Table 2 | Clinical sample test results on the HPV LAMP test versus GeneXpert in Maputo, Mozambique. Gray colored cells highlight concordance for the paired results**

| LAMP HPV | GeneXpert HPV (Reference Test) | | | | |
| --- | --- | --- | --- | --- | --- |
| | HPV16 | HPV18/45 | HPV16 & HPV18/45 | HPV16/18/45 negative | Total |
| HPV16 | 38 | 0 | 0 | 0 | 38 |
| HPV18/45 | 0 | 58 | 2 | 1 | 61 |
| HPV16 & HPV18/45 | 1 | 0 | 5 | 0 | 6 |
| HPV16/18/45 negative | 4 | 8 | 1 | 73 | 86 |
| Total | 43 | 66 | 8 | 74 | 191 |

4 years. Therefore, having gone through various processes and an extended storage time, it is possible that some sample degradation has occurred in this sample bank. In contrast, the Houston samples were collected into 350 µl PBS or nuclease-free water, which means they were ~30% more concentrated than the Mozambique samples. For patients with very low viral loads, collecting cervicovaginal swabs into smaller volumes would produce a more concentrated sample and could lead to more accurate results. Additionally, the Houston samples were only stored for up to six months before being tested on the LAMP assay, so sample degradation was likely minimal. Overall, the Houston samples are more representative of an ideal sample collection procedure for a point-of-care test, for which a cervicovaginal swab could be stored dry for future short-term use or could be collected directly into a small volume of lysis buffer for immediate testing.

We observed that the Mozambique samples that were falsely negative on the LAMP test generally contained less HPV DNA, which could be the result of HPV DNA degradation or the effects of storage in alcohol on the ability to release the HPV DNA from the stored cells[27]. For all falsely negative samples, the gDNA cellular control did amplify, affirming the adequacy of the sample and validity of reagents. It is unsurprising that some samples with low amounts of HPV DNA in a highly complex sample matrix were detectable on GeneXpert but not on the LAMP test. The input of crude sample lysate directly into the LAMP reaction introduces complexities, including the presence of cellular debris, blood and mucus, and residual NaOH, the combination of which likely contributed to some false negative results. GeneXpert avoids this issue by employing sonication for sample lysis and incorporating filtration and wash steps to purify the DNA, effectively isolating the genetic material from the rest of the sample[14]. For the LAMP assay workflow, we used extraction-free sample preparation to prioritize simplicity and accessibility, recognizing that this approach cannot match sensitivity obtained with DNA extraction. While extraction-free lysis does not allow for removal of inhibitors or concentration of target, it does have minimal user steps and greatly reduces the

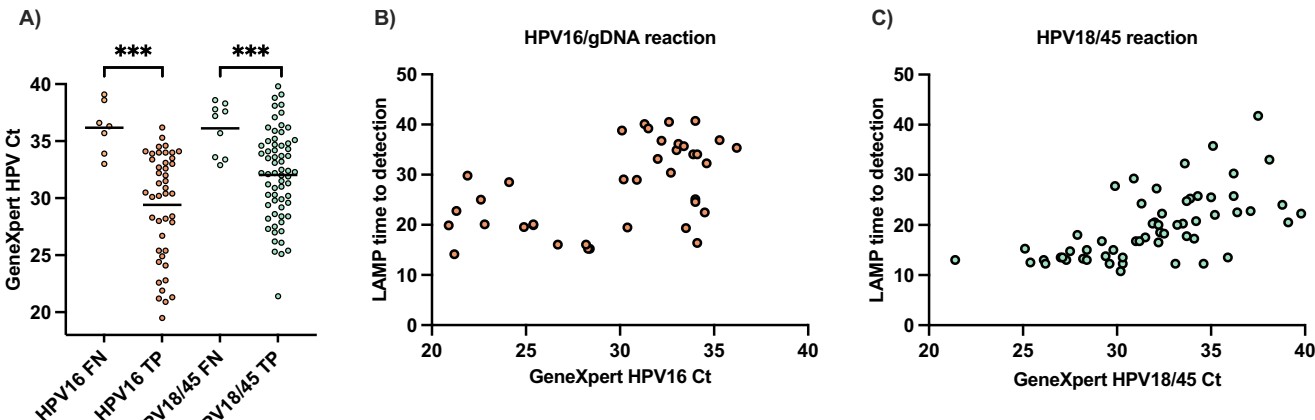

**Fig. 6 | Relationship between GeneXpert Ct values and LAMP test results.** Each data point represents results from an individual patient sample. **A** Comparison of GeneXpert Ct values for GeneXpert positive samples that were concordant by LAMP (True Positive: TP, *n* = 44 for HPV16, *n* = 65 for HPV18/45) or discordant by LAMP (False Negative: FN, *n* = 7 for HPV16, *n* = 9 for HPV18/45). Horizontal lines represent mean Ct values. The statistical significance in difference of means between true positive and false negative results was analyzed with an unpaired two-sided Welch's t-test assuming Gaussian distribution and a significance level of <0.05. Differences in mean Ct values for FN and TP are statistically significant (for HPV16 *p* < 0.0001 and for HPV18/45 *p* = 0.0004). **B** HPV16 GeneXpert Ct value versus time to detection for the HPV16/gDNA LAMP assay (*n* = 39). **C** HPV18/45 GeneXpert Ct value versus time to detection for the HPV18/45 LAMP assay (*n* = 58). Higher Ct values indicate less HPV DNA. Source data are provided as a Source Data file.

complexity of equipment required to perform the assay, significantly improving the accessibility and simplicity of the workflow. While sensitivity is maximized with DNA extraction, extraction-free sample preparation has significant potential to increase accessibility of high-performance molecular testing in low-resource settings.

Additionally, while the Mozambique samples we tested for our assay were more concentrated than the samples tested by GeneXpert, the volume of samples tested was different. GeneXpert used 1 mL of a 20 mL sample, and we used 1 µL of the same sample concentrated to 1000 µL. We opted for a small sample volume to avoid the inhibitory effects on NaOH on LAMP. Further studies are needed to evaluate if sufficient cell lysis could be obtained with lower concentrations of NaOH, which would allow for the use of a larger sample volume.

We evaluated the specificity of the LAMP assay with plasmids containing the full genome of 13 off-target HPV types in a background of gDNA and did not observe off-target amplification. Amongst the 191 Mozambique clinical samples, the only two false positive results were in the HPV18/45 LAMP assay. Interestingly, both samples were positive on the GeneXpert P3, P4, and P5 channels, all of which had low Ct values, indicating the presence of a minimum of three high-risk HPV types and high viral loads. In our specificity study, we only evaluated the presence of one off-target HPV type at a time, but considering the false positive results, it is possible that the HPV18/45 LAMP primers may produce amplification in the presence of multiple off-target HPV types. Although this may seem like a pitfall, the HPV18 and HPV45 primers used here may facilitate the expansion of this assay to include more HPV types in the future.

The WHO published its TPP for HPV screening tests for use at the point of care in 2024[5], and further developments are needed for our test to meet the requirements. We report significant progress by detecting the three HPV types that cause the majority of cervical cancer cases and an internal quality control with a one-hour extraction-free sample-to-answer workflow. The workflow time meets the TPP requirement of under one hour, so the assay could potentially be applied in a single visit screen-and-treat approach which would be particularly useful in low-resource settings, where the majority of cervical cancer cases occur. Additional requirements met by the reported test include the instrument needed for sample analysis, invalid result rate of <1%, minimal sample preparation steps, and cost

of <USD $8 (Supplementary Table 3). Additionally, all reagents and materials could easily be incorporated into a kit format.

Importantly, we demonstrated high concordance of the LAMP assay with the validated, WHO-prequalified GeneXpert HPV test. Notably, concordance and kappa values were almost identical to those reported for intra-assay reliability testing of GeneXpert, 94% and 0.88, respectively, using specimens from women attending screening and colposcopy[28], whose HPV viral load is expected to be higher than those HPV-positive samples from a screening population because the latter is enriched for cervical precancer[29,30]. And, because of those higher HPV viral loads among those with cervical precancer and cancer, we would predict that the clinical sensitivity for the LAMP assay (for HPV16/18/45-related cervical precancer and cancer) to be better than the 89.7% analytical sensitivity relative to GeneXpert as shown in this study.

To further improve this assay and meet additional TPP requirements, it is necessary to include five additional high-risk HPV types, lyophilize the LAMP reagents to eliminate pipetting steps and maximize reagent stability, test for cross reactivity with other microorganisms commonly found in the genital tract, and collect swabs directly into lysis buffer for immediate onsite testing. Additionally, usability studies in resource-limited settings should be performed to identify any barriers to implementation, assess necessary training materials, and iterate on assay design if necessary. Finally, clinical validation of the assay is required, comparing HPV results with cervical histology results from patients. Overall, this assay is sensitive, specific, and simple, making it ideal for expanding accessibility for HPV DNA testing to prevent cervical cancer in low-resource settings.

## Methods
### Ethical statement
The research presented here complies with all relevant ethical regulations. For the collection of patient samples in Houston, the study protocol was approved by the IRBs at MD Anderson Cancer Center (MDACC) (2024-0020), Harris Health System (24-05-3314), and Rice University (2024-388) prior to study initiation. Inclusion criteria for this protocol allows subjects 21 years of age or older. All participants provided written informed consent. In Mozambique, the samples were collected and banked with IRB approval from MDACC (2020–0651) and Comité Nacional de Bioética para a Saúde, Moçambique (IRB00002657). Inclusion criteria for this protocol allows subjects

30–49 years of age. All participants provided written informed consent. The samples were tested with IRB approval from Rice University (2021-48). The inclusion criteria for the protocols under which all these samples were collected stated that the subject must have a cervix. Thus, anyone without a cervix was ineligible to be screened for cervical cancer, regardless of sex or gender. For the research presented in this manuscript, no attempt was made to generalize any information about the gender of the population providing the samples, thus gender was not tracked as part of this research.

### Preparation of target DNA

DNA was extracted from SiHa, HeLa and MS751 cells with Qiagen's DNeasy Blood and Tissue Kit (Germantown, MD) according to kit instructions for cultured cells. HPV DNA was quantified via qPCR with previously published primers for the E7 gene[31] (Supplementary Table 4). For qPCR, 20 μL reactions were run with 1X PowerUp SYBR Green Master Mix (Applied Biosystems), 0.5 μM of each primer, 5 μL target DNA, and 3 μL nuclease-free water. The thermocycling protocol consisted of 50 °C for 2 min, 95 °C for 10 min, and 40 cycles of 95 °C for 10 s, 58 °C for 15 s, and 60 °C for 15 seconds. Extracted samples were serially diluted and used to determine assay limit of detection. Jurkat genomic DNA (gDNA) was purchased from ThermoFisher Scientific (Waltham, MA) (SD1111) and also serially diluted.

### LAMP reagents and reaction composition

For the DARQ LAMP HPV16, HPV18, and cellular control (gDNA) reactions, the FIP primers were labeled with a quencher and probes (Fd) were labeled with the corresponding fluorophore. For HPV45, when labeling FIP or BIP, no detectable amplification was observed, so the HPV45 LF primer was labeled instead. The HPV16 and HPV18 probes were labeled with fluorescein (FAM) and the gDNA and HPV45 probes were labeled with carboxy-X-rhodamine (ROX).

The primer sequences used in this paper can be found in Supplementary Table 4. Equal concentrations of primer and probe were mixed and heated at 95 °C for 5 min. Then, they were cooled to room temperature and added to the primer mix. The concentrations of each primer in the LAMP reaction are shown in Supplementary Table 5 and were kept consistent for every experiment. However, the percentage of labeled primer varied for different experiments and is specified for each case. The original reaction composition used for the singleplex reactions shown in Fig. 2 was based off our previous work and is presented in Supplementary Table 6. The optimized reaction compositions for the multiplexed reactions shown in Figs. 3 and 4 are presented in Supplementary Table 7. To all reactions run on the T8-ISO, one drop of mineral oil (Sigma-Aldrich; Burlington, MA; M5904) was added to the top of the reaction before capping the tube and placing in the instrument to avoid evaporation. All primers were purchased from Integrated DNA Technologies (IDT; Coralville, IA) with standard desalting, except for the primers with a fluorophore or quencher which were HPLC-purified. All LAMP reagents were purchased from New England Biolabs (NEB; Ipswich, MA), except betaine which was purchased from Sigma-Aldrich (B0300).

### Instrument settings and data processing

The majority of reactions were run on a Bio-Rad CFX Opus 96 Real-Time PCR System (Hercules, CA), which was set at a constant temperature of 65 °C with 0.705 min/cycle for up to 100 cycles, reading fluorescence every 30 s. Experiments on the Axxin T8-ISO were run at a constant temperature of 65 °C for 45 min reading fluorescence every 15 s. The FAM channel was set to a PWM of 7% and the ROX channel at 30%.

For the singleplex DARQ LAMP reactions, all fluorescence readings under 10 min were eliminated due to noise, and the first readings at 10 min were considered the baseline, which were subtracted from all other values. The positivity threshold was calculated as the average fluorescence of the NTCs plus 10 times the standard deviation. Time to

detection was determined as the time at which the fluorescence intensity exceeds the positivity threshold.

For all the multiplexed HPV16/gDNA reactions run on the Bio-Rad, all fluorescence readings under 10 min were eliminated due to noise, and the first readings at 10 min were considered the baseline, which were subtracted from all other values. A positivity threshold of 300 RFU was set manually, due to increased noise observed in the presence of multiple fluorophores in a single reaction.

For all the multiplexed HPV18/45 reactions run on the Bio-Rad, fluorescence readings were less noisy than the DARQ LAMP reactions allowing for basic data processing by the Bio-Rad software (Bio-Rad CFX Maestro 2.3 version 5.3.022.1030). Data were baseline subtracted and corrected for fluorescence drift and a single threshold per experiment was set by the Bio-Rad software.

For all HPV16/gDNA reactions run on the T8-ISO, the FAM (HPV16) channel was analyzed with a custom MATLAB (version R2022b) code. The baseline fluorescence, defined as the average fluorescence between nine and ten minutes, was first subtracted from the fluorescence signal of each sample. Cubic Hermite spline interpolation with the least squares method was then performed on the background-subtracted fluorescence signal to obtain the amplification curve. Constraints on the initial and final fluorescence values were applied, based on the observed signal, to ensure accuracy and reliability of the fitted spline. While reviewing the clinical sample data, a positivity threshold of 93.6 mV was specified to determine positive and negative samples and to determine time to detection. This threshold was applied to all other HPV16/gDNA experiments on the FAM channel run on the T8-ISO.

For all HPV16/gDNA reactions on the ROX channel and all the HPV18/45 reactions run on the T8-ISO, the average fluorescence between nine and ten minutes was specified as the baseline which was subtracted from all other data points. While reviewing the clinical sample data, positivity thresholds of 100 mV and 500 mV were set for gDNA and HPV18/45, respectively. These thresholds were applied to all other relevant experiments run on the T8-ISO.

### Limit of detection experiment with cultured cells

SiHa (HTB-35), HeLa (CCL-2), MS751 (HTB-34), and C33A (HTB-31) cells were purchased from American Type Culture Collection (ATCC; Manassas, Virginia) and passaged up to seven times before being harvested for cell counting. All cell types were pelleted and resuspended in nuclease-free water to avoid any inhibition in the LAMP reaction and then counted on a Sony MA900 Multi-Application Cell Sorter (Sony Biotechnology Inc.; San Jose, California) with a 130 μm sorting chip (LE-C3213). HPV-infected cells were counted in a background of C33A cells in different proportions to obtain the desired number of HPV-infected cells per reaction (specified in Fig. 5), maintaining the total number of cells counted per tube by the cell sorter at $2.532 \times 10^4$ cells in ~200 μL. To more realistically represent a clinical sample, we added an additional amount of C33A cells to each LAMP reaction. These were counted on a hemocytometer and diluted to approximately $10^6$ cells/mL, which was equivalent to adding an additional $10^3$ C33A cells were per LAMP reaction.

### Obtaining clinical samples

At MD Anderson Cancer Center (MDACC) and Lyndon B. Johnson Hospital (Houston, TX), one cervicovaginal swab sample per patient was collected by a healthcare provider and placed into 20 mL of PreservCyt buffer. A second sample was either self-collected or provider collected and placed into 350 μL PBS or nuclease-free water (Supplementary Tables 1 and 2). Both samples were used for research purposes only. Samples were de-identified and sent to Rice University by courier. Samples were then stored at 4 °C and used for testing within six months of collection. One milliliter of the PreservCyt sample was used for GeneXpert HPV testing and 20 μL of the PBS or water sample were used for LAMP testing. These results were used for research purposes only.

In Mozambique, cervicovaginal swab samples were collected into 20 mL PreservCyt buffer, and 1 mL was tested on GeneXpert HPV. The remaining sample was pelleted and resuspended in ~1 mL PreservCyt and stored at −20 °C for future testing. Samples were stored between 1 and 4 years before LAMP testing.

### Clinical sample testing

For the clinical samples in Houston, Texas, 2 µL of 1 M NaOH were added to a 20 µL aliquot of sample, incubated at room temperature for 10 min, and 1 µL of sample was added to each LAMP reaction. For the clinical samples in Maputo, Mozambique, 10 µL of 1 M NaOH were added to a 100 µL aliquot of sample, incubated at room temperature for 10 min, and 1 µL of sample was added to each LAMP reaction. One drop of mineral oil was added to each reaction before being incubated on the T8-ISO for 45 min.

### Statistics and reproducibility

In this study, experiments were not randomized, and investigators were not blinded to allocation during experiments and outcome assessment. For optimization experiments, replicates of the same sample were tested. We analyzed if there was a statistically significant difference in the mean time to detection in the absence and presence of different concentrations of additives. We performed an unpaired two-sided Student's t-test assuming Gaussian distribution and a significance level of <0.05 using Graphpad Prism version 10.4.1.

To determine the limit of detection with extracted DNA in the multiplexed reactions, each assay was run three separate times on the Bio-Rad. A different sample was serially diluted each time, and three replicates of each concentration of sample were tested. To determine the limit of detection with whole cells, each assay was run once on the Bio-Rad, and one sample was serially diluted for each target. Ten replicates of each concentration of sample were tested. Probit analysis was performed with AAT Bioquest's Probit Regression Calculator[32] and the probability was set to 95%.

For clinical samples, each sample was tested once, no statistical method was used to predetermine sample size, and reference test results were known before LAMP testing. One sample evaluated in Mozambique produced an oscillatory fluorescence pattern that did not resemble amplification or a negative result and was omitted from analysis. We calculated the 95% binomial exact confidence interval for concordance between the LAMP test and the reference test. We analyzed the statistical significance in difference of means between true positive and false negative clinical sample results. We performed an unpaired two-sided Welch's t-test assuming Gaussian distribution and a significance level of <0.05 using Graphpad Prism version 10.4.1.

### Ethics and inclusion statement

This work was performed by a group of researchers, clinicians, and nurses based in the United States and Mozambique. The Mozambique samples used in this study were collected as part of a larger study (ClinicalTrials.gov: NCT05359016) in which 9014 women underwent cervical cancer screening with GeneXpert HPV testing between January 2020 and January 2023[12]. Women who tested positive for HPV were followed with visual inspection with acetic acid and received treatment when clinically indicated. These samples were integral to the analytical validation of the assay presented in this work, in which we demonstrated the feasibility of its implementation in Mozambique. Funding, management, and implementation of this project was a collaborative effort and is reflected in authorship and acknowledgements.

### Reporting summary

Further information on research design is available in the Nature Portfolio Reporting Summary linked to this article.

## Data availability

All data generated in this study have been deposited in the Zenodo database under accession code https://doi.org/10.5281/zenodo.15865408[33]. Source data are provided with this paper.

## Code availability

The code used in this study to analyze the FAM channel for the HPV16/gDNA reactions run on the T8-ISO has been deposited in the Zenodo database under accession code https://doi.org/10.5281/zenodo.15930930[34].

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

## Acknowledgements

We would like to thank Dr. Jennifer Carns and Dr. Richard Schwarz (Rice University, Richards-Kortum Lab) for their administrative contributions making this work possible. We also thank Harshavardhan Deshmukh for his assistance using the cell sorter (Rice University, Bioengineering Shared Equipment Authority). We would also like to thank Cindy Melendez, Andrew Kim, Keiry Paiz, Carl Rhodes, and Ana Lopez at MD Anderson Cancer Center and LBJ Hospital who supported this work through study enrollment and sample collection. We also gratefully acknowledge the doctors and nurses from MULHER Study Team in Mozambique who supported this work through study enrollment and sample collection. Lastly, we thank the HPV Plasmid Repository within the Chronic Viral Diseases Branch at the Centers for Disease Control and Prevention, Atlanta, GA for providing the HPV plasmids used in this study. National Cancer Institute of the National Institutes of Health Award Number U01CA292741 (PIs: R.R.R.K., C.L., J.R.M.; co-investigator: M.P.S.). National Institute of Biomedical Imaging and Bioengineering of the National Institutes of Health Award Number U54EB034652 (PI: R.R.R.K.; co-investigators: C.L., J.R.M., M.P.S., and K.M.S.). The content is solely the responsibility of the authors and does not necessarily represent the official views of the National Institutes of Health. Additional support was provided from the United States National Cancer Institute through the MD Anderson Cancer Center Support Grant P30CA016672 (M.P.S., K.M.S., J.R.M., E.S.B., and M.G.) and Institutional Research Grant, The University of Texas MD Anderson Cancer Center 128997 (PI: M.P.S.; co-investigator: R.R.R.K.; collaborator: J.R.M.).

## Author contributions

Conceptualization: M.J.B., A.F.W., M.P.S., K.M.S., R.R.R.K. Methodology: M.J.B., A.F.W., P.E.C., and R.R.R.K. Software: K.G., A.E.M., and M.J.B. Investigation: M.J.B., A.F.W., and A.E.M. Resources: H.A., G.T., N.M.R.B.O., M.G., and E.H.L. Data curation: M.J.B., and A.F.W. Writing – original draft: M.J.B. and R.R.R.K. Writing – review and editing: M.J.B., A.F.W., P.E.C., J.R.M., M.P.S., K.M.S., and R.R.R.K. Visualization: M.J.B. Supervision: C.L., E.S.B., M.P.S., K.M.S., and R.R.R.K. Project administration: E.H.L., E.S.B., M.P.S., K.M.S., and R.R.R.K. Funding acquisition: C.L., M.G., J.R.M., E.S.B., M.P.S., K.M.S., and R.R.R.K.

## Competing interests

P.E.C. has received HPV tests and assays from Cepheid and Atila Biosystems at no or reduced cost for research purposes only. The remaining authors declare no competing interests.
