## [Transparent Peer Review file · Nature Communications]

One-hour extraction-free loop-mediated isothermal amplification HPV DNA assay for point-of-care testing in Maputo, Mozambique

Corresponding Author: Dr Rebecca Richards-Kortum

Version 0:

Reviewer comments:

Reviewer #1

(Remarks to the Author)

Currently, the market for HPV tests is dominated by PCR based methods. The most used HPV tests by volume globally rely on technology platforms not suitable for low resource settings by price, training level required, service level required and infrastructure required. On the other hand, a host of new simple PCR based assays that can be run on simple bench top instruments are also available. More than 1000 HPV tests have been counted by Poljak et al in their recent update of the HPV assay field. 79% has no sort of supportive evidence to detail the assays actual performance. This latter point is essential. Because without proper validation data, assay performance is as likely to harm women than to help prevent cancer.

In this context this manuscript details a simple loop mediated ITA HPV assay detecting HPV16, 18 and 45 within a one hour timeframe.

Major concerns.

The manuscript reads well. However, it is also a very long read. The authors mix basic introduction, result and materials into one very long result section detailing each step of the way. The majority of the result section should be refocused to present the main results short and concise. Current edition is more like a step-by-step manual. Also the authors blend in common knowledge in the result section that is either better suited in the intro or discussion. Example is found in lines 132-133 where the reader is tutored to an understanding about how different fluorophores enable distinction between amplicons. Assume that your readers are well versed in molecular biology.

The entire result section on optimization from single plex to multiple should be a supplementary text. Not a results section. It adds little to the main objective.

The authors fail to argue why they chose HPV 18 and 45 as targets. By absolute numbers, HPV16, 31 and 18 should be the targets if optimal cancer preventive effect is sought. Please add the main argumentation for choosing two mainly adenocarcinoma HPV gt over the more common SCC HPV31.

Also, the detection LOD of 50 SiHa cells is all fine and good. However, the authors in this exercise of analytical development lose sight of the complexities of test dynamics to the point where Table 2 lacks concordance and kappa values. Standards of the field. Also, I would have preferred if the authors validated against known proficiency references from the WHO LabNetwork Reference laboratory (Karolinska University Hospital, Stockholm, Sweden)

Moreover, positive and negative predictive value is misinterpreted for positive and negative agreement between "gold" standard and new test. There is nothing predictive in detection of artificial samples or samples evaluated on HPV detection alone. Predictive value can only be calculated if confirmed histologically endpoints are assessed with respect to test parameters.

Choice of "gold" standard is interesting and disconnected from current evidence based literature. The Cepheid assay may be a cassette based "black box" assay most hawked in low resource settings, however it needs broader validation for precision for cervical screening purposes. I refer you to the current list of validated assays for cervical screening by Arbyn et al- 2024. By effect, the comparisons made is against a non-gold standard. I suggest that the authors instead frame the cross validation as "against an assay that by its performance characteristics allow for low resource use".

Finally, I disagree with the authors who seem to seek justification for choice of instruments present 22 kg for a Biorad instrument at a cost of 20k + USD. PCR cyclers come in many sizes and cost groups and even versions that run pretty nice

on battery. And validation of HPV detection to HPV detection is analytical. Not clinical. The authors needs the final case. The performance against CIN2+ detected histology. Given the author group, this final argument is well known to them!

Overall, I am very sympathetic to the case the authors present. However, the current state of the manuscript is overreaching. I propose major revision.

Reviewer #2

(Remarks to the Author)
Please see attached.

Version 1:

Reviewer comments:

Reviewer #1

(Remarks to the Author)
Thank you for diligently answering the our questions.

Reviewer #2

(Remarks to the Author)
All key review concerns have been well addressed in the revision and it is recommended for publication at this stage.

Reviewer #1 (Remarks to the Author):

Overview Currently, the market for HPV tests is dominated by PCR based methods. The most used HPV tests by volume globally rely on technology platforms not suitable for low resource settings by price, training level required, service level required and infrastructure required. On the other hand, a host of new simple PCR based assays that can be run on simple bench top instruments are also available. More than 1000 HPV tests have been counted by Poljak et al in their recent update of the HPV assay field. 79% has no sort of supportive evidence to detail the assays actual performance. This latter point is essential. Because without proper validation data, assay performance is as likely to harm women than to help prevent cancer. In this context this manuscript details a simple loop mediated ITA HPV assay detecting HPV16, 18 and 45 within a one hour timeframe.

Major concerns. The manuscript reads well. However, it is also a very long read. The authors mix basic introduction, result and materials into one very long result section detailing each step of the way. The majority of the result section should be refocused to present the main results short and concise. Current edition is more like a step-by-step manual.

We appreciate the reviewer's perspective on the readability of the results section. We added sub headings that summarize the main results for readers who are moving quickly. We provide a concise summary after each heading and condensed the text.

Also the authors blend in common knowledge in the result section that is either better suited in the intro or discussion. Example is found in lines 132-133 where the reader is tutored to an understanding about how different fluorophores enable distinction between amplicons. Assume that your readers are well versed in molecular biology.

We removed common knowledge from the results section and relocated the information to the intro or discussion as applicable. Specifically, we removed the discussion of fluorophores (lines 132-133) that the reviewer provided as an example. Additionally, we changed statements regarding production of fluorescence during amplification and functionality of fluorophores and quenchers.

The entire result section on optimization from single plex to multiple should be a supplementary text. Not a results section. It adds little to the main objective.

The text regarding multiplex assay optimization was moved into the supplementary material.

The authors fail to argue why they chose HPV 18 and 45 as targets. By absolute numbers, HPV16, 31 and 18 should be the targets if optimal cancer preventive effect is sought. Please add the main argumentation for choosing two mainly adenocarcinoma HPV gt over the more common SCC HPV31.

The authors appreciate the concern regarding how the HPV targets were chosen. HPV 16, 18, and 45 are the most common types in SCC and in cervical cancer generally (60%, 15%, and 5% of SCC cases, and 55.8%, 14.3%, and 4.8% total cervical cancer cases, respectively)¹. The prevalence of HPV31 in total cervical cancer cases is only 3.5%. Thus, the assay described in the paper targets the three most common types of HPV in cervical cancer. In the introduction we state:

“In this work, we describe the development and evaluation of an extraction-free sample-to-answer loop-mediated isothermal amplification (LAMP) assay designed to detect the top three HPV types attributable to cervical cancer (HPV16, HPV18, and HPV45), which cause 75% of cases¹, along with a cellular control to ensure sample adequacy.”

Our long-term goal is to meet the WHO’s target product profile (TPP) for HPV screening tests for use at the point of care, the preferred requirements of which state at least a two-signal output:

- 1) carcinogenic group 1a (HPV16) individually or grouped with carcinogenic group 1b (HPV18 and HPV45; pooled or individual)
- 2) carcinogenic group 1c (33, 58, 31, 52, 35) and carcinogenic group 1d (59, 39, 51, 56) pooled.

The test we describe detects carcinogenic groups 1a and 1b and pools HPV18 and 45, in alignment with the TPP. Discussion of the future steps needed to translate this proof-of-principle test into a test suitable for population screening, including the need to add group 1c and 1d HPV types, has been added to the discussion, and is reproduced here for convenience:

“To further improve this assay and meet additional TPP requirements, it is necessary to include five additional high-risk HPV types, lyophilize the LAMP reagents to eliminate pipetting steps and maximize reagent stability, test for cross reactivity with other microorganisms commonly found in the genital tract, and collect swabs directly into lysis buffer for immediate onsite testing. Additionally, usability studies in resource-limited settings should be performed to identify any barriers to implementation, assess necessary training materials, and iterate on assay design if necessary. Finally, clinical validation of the assay is required, comparing HPV results with cervical histology results from patients. Overall, this assay is sensitive, specific, and simple, making it ideal for expanding accessibility for HPV DNA testing to prevent cervical cancer in low-resource settings.”

Also, the detection LOD of 50 SiHa cells is all fine and good. However, the authors in this exercise of analytical development lose sight of the complexities of test dynamics to the point where Table 2 lacks concordance and kappa values. Standards of the field.

We appreciate the need to report concordance with kappa values. We calculated concordance and kappa values for the Mozambique samples for HPV16, HPV18/45, and noted overall positive agreement. These values were added to the results section, and the data used to calculate them are shown below:

HPV16:

Concordance = 96.34%

kappa = 0.902

95% confidence interval: 0.831 – 0.973

LAMP	GeneXpert	
	Pos	Neg
Pos	44	0
Neg	7	140

HPV18/45:

Concordance = 94.24%

kappa = 0.877

95% confidence interval: 0.806 – 0.947

LAMP	GeneXpert	
	Pos	Neg
Pos	65	2
Neg	9	115

Overall:

Concordance = 92.67%

Overall agreement: kappa = 0.850

95% confidence interval: 0.775 – 0.925

LAMP	GeneXpert	
	Pos	Neg
Pos	104	1
Neg	13	73

Also, I would have preferred if the authors validated against known proficiency references from the WHO LabNetwork Reference laboratory (Karolinska University Hospital, Stockholm, Sweden).

Because our assay is designed to accept crude cell lysate from clinical samples rather than extracted DNA, we believe it is important to validate performance both with cells as well as with known proficiency references. The proficiency references from the WHO LabNetwork Reference laboratory contain purified plasmid DNA in a background of human genomic DNA. Evaluating performance with this type of sample alone does not test the performance impact of sample preparation (which includes cell lysis). To validate the complete assay, including extraction-free sample preparation, requires determining the limit of detection accounting for the lysis step and the impact of crude lysate on amplification. Thus, we characterized the limit of detection using cultured cells (e.g., SiHa cells isolated from uterine tissue infected with HPV16). This experimental

design is consistent with data presented for WHO prequalification for tests that have been pre-qualified, including for GeneXpert² and cobas 4800 HPV³. To make this rationale clearer, we changed the header of the cell experiment to read, “Limit of detection with extraction-free crude lysate from cultured cells.”

Importantly, we reported the LOD both for the number of whole cells, as well as for the number of copies of the relevant HPV types integrated into the human genome. These results are included in Figures 3 and 4 (Figure 3, which shows the HPV16/gDNA reaction is reproduced below).

Fig. 3. Time to detection of optimized multiplexed HPV16/gDNA reaction on Bio-Rad thermocycler and benchtop T8-ISO. A range of concentrations of SiHa (HPV16) DNA and gDNA were used as targets to evaluate the limit of detection of the reaction. The reaction was performed on the Bio-Rad (A&B) for probit analysis (n=9) and was read on the **A)** FAM channel for HPV16 detection and **B)** ROX channel for gDNA detection. The reaction was also performed on the benchtop T8-ISO (C&D), also read on the **C)** FAM channel for HPV16 detection and **D)** ROX channel for gDNA detection (n=3). Bar graphs represent mean time to detection and error bars represent standard deviation.

As requested, we also tested our assay with the known proficiency reference from the WHO LabNetwork Reference laboratory (HPV16 plasmid) to demonstrate similar performance as with extracted SiHa DNA. The results are shown below:

Amplification curves and time to detection of optimized multiplexed HPV16/gDNA reaction on Bio-Rad thermocycler with HPV16 plasmid from the WHO LabNetwork Reference laboratory. A range of concentrations of HPV16 plasmid were used as targets and results were consistent with results shown in Fig. 3. Two out of three replicates amplified for 50 input copies and consistent amplification was observed for all replicates at higher concentrations. Bar graphs represent mean time to detection and error bars represent standard deviation. n=3.

Moreover, positive and negative predictive value is misinterpreted for positive and negative agreement between “gold” standard and new test. There is nothing predictive in detection of artificial samples or samples evaluated on HPV detection alone. Predictive value can only be calculated if confirmed histologically endpoints are assessed with respect to test parameters.

We eliminated the term “predictive value” and calculate agreement instead. As shown above, we added kappa values reporting this agreement by HPV type. In the results section, we state:

“For the Mozambique samples, there was 93% concordance (177/191) between the two tests with a very good kappa value (kappa=0.851, 95% CI: 0.777 – 0.926). There was 96% concordance for HPV16 (184/191), which had the highest kappa value (kappa=0.902, 95% CI: 0.831 – 0.973). For HPV18/45 there was 95% concordance (182/192) with a kappa value of 0.888 (95% CI: 0.820 – 0.955).”

Choice of “gold” standard is interesting and disconnected from current evidence based literature. The Cepheid assay may be a cassette based “black box” assay most hawked in low resource settings, however it needs broader validation for precision for cervical screening purposes. I refer you to the current list of validated assays for cervical screening by Arbyn et al- 2024. By effect, the comparisons made is against a non-gold standard. I suggest that the authors instead frame the cross validation as “against an assay that by its performance characteristics allow for low resource use”.

Xpert HPV is one of only four high-risk HPV tests for cervical cancer screening that have received WHO prequalification⁴. Arbyn et al consider Xpert HPV suitable for

primary cervical cancer screening in their evaluation⁵. Additionally, various studies have demonstrated comparability of Xpert HPV with other validated tests^{6,7}. However, we replaced the term “gold standard” with “reference test” throughout the manuscript.

Finally, I disagree with the authors who seem to seek justification for choice of instruments present 22 kg for a Biorad instrument at a cost of 20k + USD. PCR cyclers comes in many sizes and cost groups and even versions that run pretty nice on battery.

We addressed the lack of commentary on low-cost thermocyclers with the following edits to the discussion:

“Throughout this work, we tested our reaction on two different instruments - a high complexity thermocycler (Bio-Rad CFX Opus 96 Real-Time PCR System) and a simple fluorimeter/heat block (Axxin T8-ISO) and observed no significant differences in results obtained between the two. We optimized the assays on the Bio-Rad, which provides higher throughput, allowing us to test various conditions simultaneously and implemented the assays with clinical samples on the T8-ISO. The Bio-Rad is large (33x56x36 cm and 22 kg), expensive (~ USD \$25,000), requires an electrical outlet, and yearly maintenance, making it more difficult to implement in resource-limited settings. In contrast, the T8-ISO produces the same results in a low-throughput format, is portable (18.5x20.3x11.0 cm and 2.3 kg), relatively inexpensive (~ USD \$6,500), and can be run on a battery, making it ideal for resource-limited settings. *Although we present results only on these two devices, a variety of mini thermocyclers and benchtop fluorescence isothermal readers are available on the market that could be used at the point of care. We envision this LAMP assay to be compatible with a wide assortment of devices, expanding accessibility.*”

And validation of HPV detection to HPV detection is analytical. Not clinical. The authors needs the final case. The performance against CIN2+ detected histology. Given the author group, this final argument is well known to them!

We appreciate the reviewer’s comments and agree that clinical validation against CIN2+ histology is an essential component of assay development. We acknowledge that our current study provides an analytical, rather than clinical, validation of HPV detection.

As this work represents an initial evaluation of a novel assay in a low-resource setting where histopathology infrastructure is limited, we focused on establishing analytical performance. HPV is a highly sensitive biomarker for cervical cancer^{8,9}, and HPV detection has been endorsed as the preferred screening strategy by the WHO¹⁰ and the American Cancer Society¹¹.

Our goal is to develop an assay with analytical sensitivity comparable to WHO-prequalified PCR-based tests, such as GeneXpert, but optimized for cost and workflow constraints to enable implementation in regions like Mozambique, where HPV-based screening is currently unavailable and visual inspection with acetic acid (VIA) remains standard.

Given this context, we prioritized demonstrating analytical sensitivity using clinical samples from Mozambique. Unfortunately, histology data were not available for these samples as the patients underwent a “Screen & Treat” approach per the WHO guidelines for low-resource settings where pathology services are not consistently available. We acknowledge the trade-off between evaluating the assay in our intended deployment setting versus one where full clinical validation is feasible. Throughout the revised manuscript, we clarify that we evaluated our assay analytically with clinical samples.

To partially address this limitation, we included available histology data from a small number of samples collected in Houston. While this dataset is insufficient for formal clinical validation, it is notable that four out of five samples positive for HPV16 or HPV18/45 were associated with histologically confirmed CIN2+ lesions and none of the 14 other samples were associated with CIN2+ lesions. It is worth noting that the concordance was extremely high, despite the use of banked specimens, and we would expect that concordance to be higher still on fresh specimens from women who have high-grade disease, who will have on average significantly higher HPV viral loads. These data are incorporated in the revised manuscript, and we updated the discussion to emphasize that future work will include comprehensive clinical validation using samples with available histopathological outcomes with a broader range of HPV types to meet the WHO TPP specifications.

Table S1. Comparison of GeneXpert and Histology Results for Samples Collected in Houston with Histology Results Available

Sample number	GeneXpert	LAMP	Histology	Provider/self collected	Collection buffer
1	neg	neg	Normal	Provider	PBS
2	neg	neg	Normal	Provider	PBS
3	neg	neg	Normal	Provider	PBS
4	neg	neg	Normal	Self	PBS
5	neg	neg	Normal	Provider	PBS
6	neg	neg	Normal	Provider	PBS
7	neg	neg	CIN1	Provider	PBS
8	neg	neg	CIN1	Provider	PBS
9	neg	neg	CIN1	Self	PBS
10	neg	neg	CIN1	Provider	PBS
11	neg	neg	CIN1	Self	Water
12	P3	neg	Normal	Provider	PBS
13	P5	neg	Normal	Provider	PBS
14	P4	neg	CIN1	Provider	PBS
15	HPV16	HPV16	VAIN2/3*	Provider	PBS
16	HPV16	HPV16	CIN2	Provider	PBS
17	HPV16	HPV16	CIN2/3	Self	PBS
18	HPV18/45, P3	HPV18/45	CIN1	Self	Water
19	HPV18/45	HPV18/45	CIN3	Self	Water

*Clinical note: "Fragments of endocervical and squamous epithelium with metaplastic changes"

Table S2. Comparison of GeneXpert and Histology Results for Samples Collected in Houston at Visits Where Histology Was Not Performed

Sample number	GeneXpert	LAMP	Histology	Provider/self collected	Collection buffer
20	P4/P5	neg	N/A	Provider	PBS
21	P3	neg	N/A	Provider	PBS
22	neg	neg	N/A	Provider	PBS
23	neg	neg	N/A	Provider	PBS
24	neg	neg	N/A	Provider	PBS
25	neg	neg	N/A	Provider	PBS
26	neg	neg	N/A	Provider	PBS
27	neg	neg	N/A	Provider	PBS
28	neg	neg	N/A	Provider	PBS
29	neg	neg	N/A	Provider	PBS
30	neg	neg	N/A	Provider	PBS
31	neg	neg	N/A	Provider	PBS
32	neg	neg	N/A	Provider	PBS
33	neg	neg	N/A	Provider	PBS
34	neg	neg	N/A	Provider	PBS
35	HPV16	HPV16	N/A	Provider	PBS
36	HPV16	HPV16	N/A	Provider	PBS
37	HPV18/45	HPV18/45	N/A	Provider	PBS
38	HPV18/45	HPV18/45	N/A	Provider	PBS

Overall, I am very sympathetic to the case the authors present. However, the current state of the manuscript is overreaching.

I propose major revision.

We thank the reviewer for their positive impression of our work and have modified the manuscript as detailed above to remove imprecise language that contributed to the impression of overreaching.

References

1. International Agency for Research on Cancer (IARC). *Cervical Cancer Screening*. vol. 18.
2. Public report for Xpert HPV (PQDx 0268-070-00) | WHO - Prequalification of Medical Products (IVDs, Medicines, Vaccines and Immunization Devices, Vector Control). <https://extranet.who.int/prequal/WHOPR/public-report-xpert-hpv-pqdx-0268-070-00-0>.
3. Public Report for cobas 4800 HPV Test, (PQDx 0466-046-00) | WHO - Prequalification of Medical Products (IVDs, Medicines, Vaccines and Immunization Devices, Vector Control). <https://extranet.who.int/prequal/WHOPR/public-report-cobas-4800-hpv-test-pqdx-0466-046-00>.
4. WHO prequalifies additional HPV test, expanding options as countries pursue cervical cancer elimination. <https://www.who.int/news/item/14-06-2023-who-prequalifies-additional-hpv-test-expanding-options-as-countries-pursue-cervical-cancer-elimination>.
5. Arbyn, M. *et al.* 2020 list of human papillomavirus assays suitable for primary cervical cancer screening. *Clin. Microbiol. Infect.* **27**, 1083–1095 (2021).
6. Rabaan, A. A., Alfaraj, S. A. & Alkhalifah, M. A. Comparison of the Cepheid Xpert HPV test and the HC2 High-Risk HPV DNA Test for detection of high-risk HPV infection in cervical smear samples in SurePath preservative fluid. *J. Med. Microbiol.* **67**, 676–680 (2018).
7. Einstein, M. H. *et al.* Clinical Evaluation of the Cartridge-Based GeneXpert Human Papillomavirus Assay in Women Referred for Colposcopy. *J. Clin. Microbiol.* **52**, 2089–2095 (2014).

8. Benevolo, M. *et al.* Sensitivity, Specificity, and Clinical Value of Human Papillomavirus (HPV) E6/E7 mRNA Assay as a Triage Test for Cervical Cytology and HPV DNA Test
▽ . *J. Clin. Microbiol.* **49**, 2643–2650 (2011).
9. Ramírez, A. T. *et al.* Performance of cervical cytology and HPV testing for primary cervical cancer screening in Latin America: an analysis within the ESTAMPA study. *Lancet Reg. Health – Am.* **26**, (2023).
10. *WHO Guideline for Screening and Treatment of Cervical Pre-Cancer Lesions for Cervical Cancer Prevention.* (World Health Organization, Geneva, 2021).
11. Fontham, E. T. H. *et al.* Cervical cancer screening for individuals at average risk: 2020 guideline update from the American Cancer Society. *CA. Cancer J. Clin.* **70**, 321–346 (2020).

Reviewer #2

Review of NCOMMS-25-12470: "One-hour extraction-free loop-mediated isothermal amplification HPV DNA assay for point-of-care testing in Maputo, Mozambique" by Barra et al.

Overview

This article by Barra and colleagues describes a loop-mediated isothermal amplification (LAMP) test for HPV16, HPV18, and HPV 14 along with a human cellular control. The assay uses strand displacing quenched fluorophores on the LAMP primers that emit fluorescence upon binding to their target sequence (DARQ LAMP). The assay is performed on lysates of cervicovaginal swabs with no additional sample extraction and is run in a benchtop heater with fluorometric readout. In particular, this extraction-free lysis with NaOH is a valuable addition to previous work that then enables translation to field testing. The team then tested the rapid assay in both the US and in Mozambique and demonstrate high concordance with existing HPV tests.

Overall, this is a very well written manuscript that goes above and beyond typical bioassay engineering to demonstrate the clinical application in the field on two continents. Prior to publication a few key points should be discussed addressing how the work meets, or moves closer, to the World Health Organization target product profile for HPV detection and the competitive advantage of the assay as well as its challenges with sample volume (aspects to consider are below). A number of minor points in the discussion, methods and figures, are brought up below that could be further clarified or added/removed in order to improve the manuscript prior to its publication.

Missing Discussion Points

WHO Target produce Profile for HPV management

- a) The TPP indicates testing of minimum 8 hrHPV types (subgroups 1a, 1b, 1c): 16, 18, 45, 33, 58, 52, and 35 at the point-of-care. How does this work contribute to the advancement towards this capability (or demonstrated challenges achieving this). What else is needed to reach the TPP requirements (added subgroups, reagent stabilization, connectivity, cost, testing cross reactivity with non-HPV organisms, etc.)

(<https://iris.who.int/bitstream/handle/10665/379099/9789240100275-eng.pdf>)

We agree with the importance of discussing how our work advances toward the WHO's TPP.

Major advancements include:

- Detection of the three most oncogenic HPV types and an internal control
- Sample-to-answer time of less than one hour
- Invalid result rate of <1%
- Extraction-free sample preparation with minimal sample handling
- Use of simple and robust instrument
- Cost of < USD \$8
- Potential use as a kit

We also note that there is significant further work needed for our assay to meet the requirements of the TPP, in brief:

- Include five additional high-risk HPV types

- Lyophilize LAMP reagents
- Test for cross-reactivity with other microorganisms commonly found in the genital tract
- Evaluate necessary training materials
- Clinical validation with histology results

We added the following paragraphs to the introduction and discussion to highlight the relationship between our work and the TPP goals:

Introduction:

“In late 2024, the WHO published a target product profile (TPP) for point-of-care HPV screening tests¹, specifying minimum and preferred characteristics. Tests that meet the TPP requirements are needed to meet the WHO’s goal of screening 70% of women worldwide twice in their life between ages 35 and 45 by the year 2030. These tests are especially needed in low-resource settings. To meet the TPP requirements, tests should be able to detect at least eight high-risk HPV types that cause approximately 97% of cervical cancer cases. Tests should include an internal quality control, temperature-stable reagents, and should not cross-react with other microorganisms that can be found in the genital tract. Additionally, tests should include all consumables in a kit that is easy to use and minimizes cost and specimen handling.”

Discussion:

“The WHO published its target product profiles (TPP) for HPV screening tests for use at the point of care in 2024¹, and further developments are needed for our test to meet the published requirements. We report significant progress by detecting the three HPV types that cause the majority of cervical cancer cases and an internal quality control with a one-hour extraction-free sample-to-answer workflow. The workflow time meets the TPP requirement of under one hour, so the assay could potentially be applied in a single visit screen-and-treat approach which would be particularly useful in low-resource settings, where the majority of cervical cancer cases occur. Additional requirements met by the reported test include the instrument needed for sample analysis, invalid result rate of <1%, minimal sample preparation steps, and cost of < US \$8 (Table S1). Additionally, all reagents and materials could easily be incorporated into a kit format.

To further improve this assay and meet additional TPP requirements, it is necessary to include five additional high-risk HPV types, lyophilize the LAMP reagents to eliminate pipetting steps and maximize reagent stability, test for cross reactivity with other microorganisms commonly found in the genital tract, and collect swabs directly into lysis buffer for immediate onsite testing. Additionally, usability studies in resource-limited settings are needed to identify any barriers to implementation, assess necessary training materials, and iterate on assay design if necessary. Finally, clinical validation of the assay is required, comparing HPV results with cervical histology results from patients. Overall, this assay is sensitive, specific, and simple, making it ideal for expanding accessibility for HPV DNA testing to prevent cervical cancer in low-resource settings.”

Table S1. Cost breakdown of LAMP test

Materials	USD
Puritan Flocked Swab	\$0.36
Sample collection tube	\$0.36
Sample collection buffer	\$0.01
Reaction tubes	\$0.32
Reagents	\$4.06
Total	\$5.11

These numbers are based on the authors' current costs and could be reduced if HPV LAMP assay kits are produced at scale.

Volume of sample:

- a) Only 1 μ L of lysed sample is added to each reaction. Much of the limit of detection is due to this small sample volume (e.g. compared to 1mL of sample per GeneXpert cartridge). Could a slightly higher volume of sample per reaction (or higher total reaction volume) have improve the LoD? If not, e.g. due to inhibitory NaOH concentrations, could reducing the concentration of NaOH have provided sufficient lysis to reduce inhibition and improve total volume that could be added? Would improvements in the assay be worth the added cost?

When lysing cells with 1M NaOH, higher sample volumes would have led to inhibitory concentrations of NaOH in LAMP. We are conducting further research to determine the minimum concentration necessary for sufficient lysis, which would allow for a larger sample volume in the LAMP reaction. We do not think a higher reaction volume would be worth the cost, since we achieved 100% concordance with the samples collected in Houston. These samples were collected into water or PBS, stored for only up to 6 months, and were not concentrated or modified in any way before using.

To address this and the next comment, we added the following paragraph to the discussion regarding the Mozambique samples:

“Additionally, while the samples we tested for our assay were more concentrated than the samples tested by GeneXpert, the volume of samples tested was different. GeneXpert used 1 mL of a 20 mL sample, and we used 1 μ L of the same sample concentrated to 1000 μ L. We opted for a small sample volume to avoid the inhibitory effects on NaOH on LAMP. Further studies are needed to evaluate if sufficient cell lysis could be obtained with lower concentrations of NaOH, which would allow for the use of a larger sample volume.”

- b) The actual amount of DNA being tested by GeneXpert and by the new assay were not the same - in fact, the pelleted cells give an advantage to the novel assay (Only disadvantages of the PreservCyt storage were discussed, not the 20x sample concentration that the pelleting and storage provided).

While the samples we tested for our assay were more concentrated than the samples tested by GeneXpert, the volume of samples tested was different. GeneXpert used 1 mL of a 20 mL

sample (1 mL/20 mL) and we used 1 μ L of the same sample concentrated to 1000 μ L.

To address this and the previous comment, we added the following paragraph to the discussion:

“Additionally, while the samples we tested for our assay were more concentrated than the samples tested by GeneXpert, the volume of samples tested was different. GeneXpert used 1 mL of a 20 mL sample, and we used 1 μ L of the same sample concentrated to 1000 μ L. Overall, the number of cells in the GeneXpert sample was greater than the number of cells in the LAMP sample. We opted for a small sample volume to avoid the inhibitory effects on NaOH on LAMP. Further studies are needed to evaluate if sufficient cell lysis could be obtained with lower concentrations of NaOH, which would allow for the use of a larger sample volume.”

Competitive Advantage

- a) Concerns with cost, maintenance, and temperature control are brought up in the introduction for the GeneXpert and justification for need for a new assay. A direct comparison should be addressed in the discussion (costs of tubes, reagents, instrument, etc as well as temperature control and maintenance requirements for clinical use) or be removed as justification concerns.

We revised the statement in the introduction to directly compare the two tests: “Despite its ease of use, GeneXpert’s major limitation is its high per-test cost of approximately USD \$15 and a minimum instrument cost of USD \$9,420².”

We compare this to the T8 instrument cost (US \$6,500) and LAMP per-test cost (< US \$8) in the discussion.

- b) In addition to battery power (which other mini-thermal cyclers claim), the Axxin T8-ISO claims a competitive advantage due to the magnetic mixing capabilities. Were these used in the study? If so, please state this in the methods. If not, could other fluorescent isothermal readers be used instead and would that broaden the applicability of the assay to other labs?

The battery and mixing capabilities of the T8 were not used in this study. Mixing is not needed for LAMP. We believe most other fluorescent isothermal readers could be used, but the only other one we have tried is the AmpliFire Isothermal Fluorometer by Agdia, Inc, which worked well for our assay. The following edits have been made:

Throughout this work, we tested our reaction on two different instruments - a high complexity thermocycler (Bio-Rad CFX Opus 96 Real-Time PCR System) and a simple fluorimeter/heat block (Axxin T8-ISO) and observed no significant differences in results obtained between the two. We optimized the assays on the Bio-Rad, which provides higher throughput, allowing us to test various conditions simultaneously and implemented the assays with clinical samples on the T8-ISO. The Bio-Rad is large (33x56x36 cm and 22 kg), expensive (~ USD \$25,000), requires an electrical outlet, and yearly maintenance, making it more difficult to implement in resource-

limited settings. In contrast, the T8-ISO produces the same results in a low-throughput format, is portable (18.5x20.3x11.0 cm and 2.3 kg), relatively inexpensive (~ USD \$6,500), and can be run on a battery, making it ideal for resource-limited settings. Although we present results only on these two devices, a variety of mini thermocyclers and benchtop fluorescence isothermal readers are available on the market that could be used at the point of care. We envision our LAMP assay to be compatible with a wide assortment of devices, expanding accessibility.

Minor Discussion:

- a) Line 16 page 22: all thermal cycler instruments require “electricity” – Presumably the authors mean the need for an electrical outlet attached to the electrical grid or generator and has a wall cord. Please clarify.

The sentence now reads:

“The Bio-Rad is large (33x56x36 cm and 22 kg), expensive (~ USD \$25,000), requires an electrical outlet, and yearly maintenance, making it more difficult to implement in resource-limited settings.”

- b) Page 22 line 29: How limit of detection is defined should be included along with the confidence level rather than hedged as a range of numbers.

We repeated the experiments with extracted DNA for the multiplexed reaction with more replicates to conduct probit analysis, and the results have been added to the main text. We added the following paragraph to the methods section:

“To determine the limit of detection of HPV16, HPV18, HPV45, and gDNA in the multiplexed reactions, three technical replicates and three biological replicates were performed for each target on the Bio-Rad. Probit analysis was performed with AAT Bioquest’s Probit Regression Calculator³ and the probability was set to 95%.”

The limit of detection for extracted DNA with 95% probability for HPV16, HPV18, and HPV45, respectively, was 57, 11.7, 14.3 copies/reaction. For gDNA, the limit of detection was 0.167 ng/reaction.

For the sample-to-answer experiment with cells, we performed 10 technical replicates and one biological replicate for each cell type and the limit of detection with 95% probability for SiHa, and HeLa cells, respectively, was 30.4 and 7.6 cells/reaction. For MS751, the lowest cell count tested was 10 cells/reaction, for which all replicates amplified. Therefore, it was not possible to conduct probit analysis for MS751.

- c) Page 23 line 448, While the inhibitory effects of sample storage are important to include, the argument that PreservCyt is the cause of inhibition should be removed. At the concentrations tested (or even double that concentration), no statistically significant difference in LAMP reaction was demonstrated. More likely the NaOH was a potential inhibitor in these lysed samples.

We thank the reviewer for this observation. The PreservCyt argument has been removed.

- d) Page 24 line 467. Please provide a reference for “the small loss in sensitivity is generally seen a reasonable tradeoff given broader accessibility.” Also please define “small”.

We revised the statement to read:

“While extraction-free lysis does not allow for removal of inhibitors or concentration of target, it does reduce the device complexity requirement greatly and has minimal user steps, significantly improving the accessibility and simplicity of the workflow. While sensitivity is maximized with DNA extraction, extraction-free sample preparation has significant potential to increase accessibility of high-performance molecular testing in low-resource settings.”

- e) Page 27, line 534 “a positivity threshold of 93.6 mV was specified to determine positive and negative samples and to determine time to detection”. Was this threshold determined *a priori* or only after the data was reviewed? How would the authors envision the evaluation proceed in the future for clinical testing the value was determined after the data collection?

The threshold was determined after the data were reviewed, and this has been now clarified in the manuscript. We believe a larger number of clinical samples is needed to set a final positivity threshold. We plan to continue validating the assay with clinical samples to determine a positivity threshold that would align with the reference test for the majority of samples.

- f) The DARQ LAMP assay is stated to have been less sensitive than intercalating dyes alone for HPV 18 and 45, however HPV 16 was not evaluated with intercalating dye and has the worst limit of detection of the 3 assays. Is there potential to improve HPV 16 detection by using an intercalating dye instead (and would there be disadvantages to this with the multiplexing)?

If HPV16 was implemented with an intercalating dye, genotyping in a multiplexed format would not be possible. We would not be able to distinguish HPV16 amplification from gDNA amplification in Tube #1. HPV16 primers could be included in Tube #2 with HPV18 and 45 and intercalating dye, but again, genotyping would not be possible. This would complicate the comparisons with GeneXpert, as GeneXpert reports HPV16 results separate from HPV18/45 results. Additionally, although all hrHPV types are treated the same way in low-resource settings in a screen-and-treat approach, genotyping HPV16 is needed for a screen-triage-treat approach.

Minor Methods:

- a) The extraction-free lysis is a key novelty of the manuscript that makes it feasible to perform the assay in field. It is a shame that the description and results of this process are buried in the methods and supplemental. They should highlighted in the main manuscript.

We thank the reviewer for this comment and added the following statements to the introduction:

“Our workflow minimizes specimen handling by incorporating extraction-free sample

preparation, in which we chemically lyse cervicovaginal swab samples and add the lysate directly to the LAMP reagents. The simplicity of this step is a key factor that allows the assay to be performed in the field.”

- b) Please clarify whether the MgSO₄ (Fig S3) was optimized and kept uniform before the betaine optimization (Fig S4) (and what MgSO₄ concentration was used during betaine in the methods).

The description of the optimization process was moved to the supplementary material and was clarified there:

“Next, concentrations of magnesium sulphate (MgSO₄, Fig. S3) and betaine (Fig. S4) were optimized in the multiplexed reaction on the Bio-Rad. MgSO₄ concentration was optimized first and the selected concentration of 8 mM was used in the betaine optimization experiment. The optimal concentrations selected for the shortest time to detection and consistent amplification were 8 mM MgSO₄ and 0.4 mM betaine.”

- c) Cycle thresholds for detection are typically set at ~10 Standard Deviations above background (in PCR), however, the background fluorescence in most of these plots (e.g. S1, S3, S4, S7) is not set with a Y-axis of 0. The process should be described in the methods for samples other than just clinical samples (currently “a positivity threshold” is described only for only the clinical samples). Presumably a similar “average fluorescence between nine and ten minutes” was the baseline for these as well but it is not stated.

We revised the methods for clarification.

For the optimization experiments shown in the supplementary material (S1, S3, S4, S7), the raw data is shown (no baseline subtraction). A positivity threshold was not set for these experiments because it was not necessary to determine time to detection, only to show the effects of different concentrations of reagents on amplification and readout (off-target amplification, noise, baseline fluorescence, etc.). This has been clarified in the text.

Additionally, the “Instrument settings and data processing” section of the methods has been rewritten for clarity and greater specificity:

“The majority of reactions were run on a Bio-Rad CFX Opus 96 Real-Time PCR System (Hercules, CA), which was set at a constant temperature of 65°C with 0.705 min/cycle for up to 100 cycles, reading fluorescence every 30 seconds. Experiments on the Axxin T8-ISO were run at a constant temperature of 65°C for 45 minutes reading fluorescence every 15 seconds. The FAM channel was set to a PWM of 7% and the ROX channel at 30%.

For the singleplex DARQ LAMP reactions, all fluorescence readings under 10 minutes were eliminated due to noise, and the first readings at 10 minutes were considered the baseline, which were subtracted from all other values. The positivity threshold was calculated as the average fluorescence of the NTCs plus 10 times the standard deviation. Time to detection was determined as the time at which the fluorescence intensity exceeded the positivity threshold.

For all the HPV16/gDNA reactions run on the Bio-Rad, all fluorescence readings under 10 minutes were eliminated due to noise, and the first readings at 10 minutes were considered the baseline, which were subtracted from all other values. A positivity threshold of 300 RFU was set manually, due to increased noise observed in the presence of multiple fluorophores in a single reaction.

For all the HPV18/45 reactions run on the Bio-Rad, fluorescence readings were less noisy than the DARQ LAMP reactions allowing for basic data processing by the Bio-Rad software. Data were baseline subtracted and corrected for fluorescence drift and a single threshold per experiment was set by the Bio-Rad software.

For all HPV16/gDNA reactions run on the T8-ISO, the FAM (HPV16) channel was analyzed with a custom MATLAB code. The baseline fluorescence, defined as the average fluorescence between nine and ten minutes, was first subtracted from the fluorescence signal of each sample. Cubic Hermite spline interpolation with the least squares method was then performed on the background-subtracted fluorescence signal to obtain the amplification curve. Constraints on the initial and final fluorescence values were applied, based on the observed signal, to ensure accuracy and reliability of the fitted spline. While reviewing the clinical sample data, a positivity threshold of 93.6 mV was specified to determine positive and negative samples and to determine time to detection. This threshold was applied to all other HPV16/gDNA experiments on the FAM channel run on the T8-ISO.

For all HPV16/gDNA reactions on the ROX channel and all the HPV18/45 reactions run on the T8-ISO, the average fluorescence between nine and ten minutes was specified as the baseline which was subtracted from all other data points. While reviewing the clinical sample data, positivity thresholds of 100 mV and 500 mV were set for gDNA and HPV18/45, respectively. These thresholds were applied to all other relevant experiments.”

Minor Figures:

- a) Since instruments used for testing are changed throughout the manuscript, the legends of each figure should indicate the instrument type used for that graph/set of graphs. E.g. the use of 25% label and 7% FIP labels would benefit from clearly stating the difference in instruments

These clarifications were made to the figure legends.

- b) Would be helpful to use the same Y axes on each part of the figure (e.g. plots in S6 and S9 are all different from one another making comparisons across groups hard)

Figures S6 and S9 were updated as requested:

- c) In Fig S7: a lot of off-target HPV strains seemed to have amplified in the ROX channel – why? Please add discussion of this.

The off-target HPV plasmids were mixed with 10 ng gDNA/reaction to represent clinical samples more realistically. Therefore, amplification in the ROX (gDNA) channel was expected (it is not off-target amplification). In the results section we state:

“The specificity of both reactions was tested with plasmids containing the full genomes of 13 off-target HPV types developed by the Centers for Disease Control and Prevention 22 (Fig. S7). Additionally, a background of gDNA was added to each sample to more realistically represent clinical samples. As expected, in the HPV16/gDNA assay (Fig. S7A) all three replicates of the on-target HPV16 plasmid amplified at 10 minutes. All the samples produced amplification in the ROX channel due to the background of gDNA.”

The caption for Figure S7 was modified for clarification:

“**Fig. S7. Specificity of the multiplexed assays on the Bio-Rad thermocycler.** For each HPV type, 10^6 copies/reaction in a background of 10 ng gDNA/reaction were used. A) HPV16/gDNA reaction with n=3 for HPV, n=6 for gDNA, and n=6 for NTCs. Only the HPV16 sample amplified in the FAM channel, whereas all samples amplified in the ROX channel due to the background of gDNA. Raw data shown. B) HPV18/45 reaction with n=3 for HPV, n=3 for gDNA, and n=6 for NTCs. Data were baseline subtracted and corrected for fluorescence drift by the Bio-Rad software.”

Minor References

- a) Page 25 Line 499 describes “quantified via qPCR as previously described”. However, checking this reference leads to yet another reference. The original reference that actually describes the method should be used, not one that describes a different reference.

This part of the methods was edited as below:

“DNA was extracted from SiHa, HeLa and MS751 cells with Qiagen’s DNeasy Blood and Tissue Kit (Germantown, MD) according to kit instructions for cultured cells. HPV DNA was quantified via

qPCR with previously published primers for the E7 gene (for HPV16, primer pair two was used)⁴.

Reference 4 is:

Gao, G. et al. A novel RT-PCR method for quantification of human papillomavirus transcripts in archived tissues and its application in oropharyngeal cancer prognosis. *Int. J. Cancer J. Int. Cancer* **132**, 882–890 (2013)

References

1. World Health Organization. *Target Product Profiles for Human Papillomavirus Screening Tests to Detect Cervical Pre-Cancer and Cancer*. (World Health Organization, 2024).
2. Cepheid pricing. *FIND DxConnect Marketplace* <https://dxc-marketplace.finddx.org/pages/cepheid-accessible-pricing>.
3. Probit Regression Calculator | AAT Bioquest. <https://www.aatbio.com/tools/probit-model-regression-analysis-calculator>.
4. Gao, G. et al. A novel RT-PCR method for quantification of human papillomavirus transcripts in archived tissues and its application in oropharyngeal cancer prognosis. *Int. J. Cancer J. Int. Cancer* **132**, 882–890 (2013).

Review of NCOMMS-25-12470: “One-hour extraction-free loop-mediated isothermal amplification HPV DNA assay for point-of-care testing in Maputo, Mozambique” by Barra et al.

Overview

This article by Barra and colleagues describes a loop-mediated isothermal amplification (LAMP) test for HPV16, HPV18, and HPV 14 along with a human cellular control. The assay uses strand displacing quenched fluorophores on the LAMP primers that emit fluorescence upon binding to their target sequence (DARQ LAMP). The assay is performed on lysates of cervicovaginal swabs with no additional sample extraction and is run in a benchtop heater with fluorometric readout. In particular, this extraction-free lysis with NaOH is a valuable addition to previous work that then enables translation to field testing. The team then tested the rapid assay in both the US and in Mozambique and demonstrate high concordance with existing HPV tests.

Overall, this is a very well written manuscript that goes above and beyond typical bioassay engineering to demonstrate the clinical application in the field on two continents. Prior to publication a few key points should be discussed addressing how the work meets, or moves closer, to the World Health Organization target product profile for HPV detection and the competitive advantage of the assay as well as its challenges with sample volume (aspects to consider are below). A number of minor points in the discussion, methods and figures, are brought up below that could be further clarified or added/removed in order to improve the manuscript prior to its publication.

Missing Discussion Points

WHO Target produce Profile for HPV management

- a) The TPP indicates testing of minimum 8 hrHPV types (subgroups 1a, 1b, 1c): 16, 18, 45, 33, 58, 52, and 35 at the point-of-care. How does this work contribute to the advancement towards this capability (or demonstrated challenges achieving this). What else is needed to reach the TPP requirements (added subgroups, reagent stabilization, connectivity, cost, testing cross reactivity with non-HPV organisms, etc.)
(<https://iris.who.int/bitstream/handle/10665/379099/9789240100275-eng.pdf>)

Volume of sample:

- a) Only 1 uL of lysed sample is added to each reaction. Much of the limit of detection is due to this small sample volume (e.g. compared to 1mL of sample per GeneXpert cartridge). Could a slightly higher volume of sample per reaction (or higher total reaction volume) have improve the LoD? If not, e.g. due to inhibitory NaOH concentrations, could reducing the concentration of NaOH have provided sufficient lysis to reduce inhibition and improve total volume that could be added? Would improvements in the assay be worth the added cost?
- b) The actual amount of DNA being tested by GeneXpert and by the new assay were not the same - in fact, the pelleted cells give an advantage to the novel assay (Only disadvantages of the PreservCyt storage were discussed, not the 20x sample concentration that the pelleting and storage provided).

Competitive Advantage:

- a) Concerns with cost, maintenance, and temperature control are brought up in the introduction for the GeneXpert and justification for need for a new assay. A direct comparison should be addressed in the discussion (costs of tubes, reagents, instrument, etc as well as temperature control and maintenance requirements for clinical use) or be removed as justification concerns.
- b) In addition to battery power (which other mini-thermal cyclers claim), the Axxin T8-ISO claims a competitive advantage due to the magnetic mixing capabilities. Were these used in the study? If so, please state this in the methods. If not, could other fluorescent isothermal readers be used instead and would that broaden the applicability of the assay to other labs?

Minor Discussion:

- a) Line 16 page 22: all thermal cycler instruments require “electricity” – Presumably the authors mean the need for an electrical outlet attached to the electrical grid or generator and has a wall cord. Please clarify.
- b) Page 22 line 29: How limit of detection is defined should be included along with the confidence level rather than hedged as a range of numbers.
- c) Page 23 line 448, While the inhibitory effects of sample storage are important to include, the argument that PreservCyt is the cause of inhibition should be removed. At the concentrations tested (or even double that concentration), no statistically significant difference in LAMP reaction was demonstrated. More likely the NaOH was a potential inhibitor in these lysed samples.
- d) Page 24 line 467. Please provide a reference for “the small loss in sensitivity is generally seen a reasonable tradeoff given broader accessibility.” Also please define “small”.
- e) Page 27, line 534 “a positivity threshold of 93.6 mV was specified to determine positive and negative samples and to determine time to detection”. Was this threshold determined *a priori* or only after the data was reviewed? How would the authors envision the evaluation proceed in the future for clinical testing the value was determined after the data collection?
- f) The DARQ LAMP assay is stated to have been less sensitive than intercalating dyes alone for HPV 18 and 45, however HPV 16 was not evaluated with intercalating dye and has the worst limit of detection of the 3 assays. Is there potential to improve HPV 16 detection by using an intercalating dye instead (and would there be disadvantages to this with the multiplexing)?

Minor Methods:

- a) The extraction-free lysis is a key novelty of the manuscript that makes it feasible to perform the assay in field. It is a shame that the description and results of this process are buried in the methods and supplemental. They should highlighted in the main manuscript.
- b) Please clarify whether the MgSO₄ (Fig S3) was optimized and kept uniform before the betaine optimization (Fig S4) (and what MgSO₄ concentration was used during betaine in the methods).
- c) Cycle thresholds for detection are typically set at ~10 Standard Deviations above background (in PCR), however, the background fluorescence in most of these plots (e.g. S1, S3, S4, S7) is not set with a Y-axis of 0. The process should be described in the methods for samples other than just clinical samples (currently “a positivity threshold” is described only for only the clinical samples).

Presumably a similar “average fluorescence between nine and ten minutes” was the baseline for these as well but it is not stated.

Minor Figures:

- a) Since instruments used for testing are changed throughout the manuscript, the legends of each figure should indicate the instrument type used for that graph/set of graphs. E.g. the use of 25% label and 7% FIP labels would benefit from clearly stating the difference in instruments
- b) Would be helpful to use the same Y axes on each part of the figure (e.g. plots in S6 and S9 are all different from one another making comparisons across groups hard)
- c) In Fig S7: a lot of off-target HPV strains seemed to have amplified in the ROX channel – why? Please add discussion of this.

Minor References

- a) Page 25 Line 499 describes “quantified via qPCR as previously described”. However, checking this reference leads to yet another reference. The original reference that actually describes the method should be used, not one that describes a different reference.